# Charge symmetry breaking in neutral polyzwitterions

**Yeseul Lee** & **Murugappan Muthukumar** ✉

Response of polar and neutral monomers in macromolecules to an electric field in their crowded aqueous solutions remains as an unchartered area of research, in contrast with well understood behavior of ionized groups. Wondering whether such monomers impart merely frictional resistance or cooperate with the other ionic groups in their electrophoretic mobility, we investigate single-molecule electrophoresis of a couple of neutral polyzwitterions which have charge-neutrality with strong acidic group and permanent positive charge within their repeat units. Combining experiments and theory, we study the roles of dipole orientation of zwiterionic monomers and salt identity. Here, we report that charge-neutral polyzwitterions exhibit mobility with rectification in their direction of movement because of charge symmetry breaking arising from differential counterion bindings due to gradients in the local dielectric constant around chain backbone, and thus opening a new avenue to understand dipolar polymers of broad interest and applications.

Directed movement of macromolecules in crowded aqueous environments is ubiquitous in myriads of biological and synthetic contexts, and is essential to sustain life[1–7]. Generally, these macromolecules are constituted by sequences of monomers with diverse attributes such as ionic charge (positive and negative), electric dipole, hydrophobicity, etc[1–8]. Inevitably, movement of such chemically rich macromolecules inside crowded aqueous environments is extremely complex, due to a confluence of various electrostatic, van der Waals, and hydrodynamic forces among the solute molecules (big and small), dissociated ions, and water.

Stoked by continuing thirst for a fundamental understanding of motion of macromolecules, starting from the pioneering work of Einstein[9], and by the societal need to separate, characterize, and sequence the various biological polymers, there has been a tremendous progress in this field of study[10–47]. The principal experimental method in these studies is electrophoresis of macromolecules under an external electric field in a variety of media that include hydrogels, porous membranes, and protein- and solid-state-naopores[39]. So far, the focus in these experiments has primarily been on uniformly charged (positive and negative) polyelectrolytes.

Although there have been studies on the electrophoretic translocation of weakly charged proteins[19,33,43,46] and polyethylene glycol

(considered neutral but acquiring polycation behavior in solutions at high ionic strength)[25,31,34], the contribution from electrically neutral zwitterionic moieties to electrophoretic mobility has so far been unexplored. Here, we consider polyzwitterions as neutral based only on their chemical structure's electrical charges determined by protonation/deprotonation equilibria using titration curves. Unlike normal neutral polymers without charged functional groups, the neutral polyzwitterions exhibit unique properties such as the anti-polyelectrolyte effect and anti-fouling functions, which make them significant from fundamental and practical points of view[48–53]. The natural question that arises is whether such neutral groups are electrically silent and become a burden for other charges in the molecule to move forward, or they also electrically contribute and regulate the molecule's movement.

Central to the mobility of charged macromolecules is the charge of the individual charged groups of the molecule. In solutions, the native degree of ionization (obtained from titration curves, for example) is modified into an effective degree of ionization (and inhomogeneous charge distribution on the molecule) due to omnipresent binding of the ionic group with its corresponding counterion in the solution. The resultant effective charge of the ionic group arising from such counterion condensation is dictated by the ion-binding

Department of Polymer Science and Engineering, University of Massachusetts, Amherst, MA, USA. ✉e-mail: muthu2346@gmail.com

energy, $\Delta G = -e^2/(4\pi\bar{\epsilon}\epsilon_\ell r)$ ($e$ is the electronic charge, $\bar{\epsilon}$ is permittivity of vacuum, $\epsilon_\ell$ is the local dielectric constant in the vicinity of ion-pair formation, and $r$ is the charge separation distance in the ion-pair) for monovalent ions. In all theories of counterion condensation composed so far[10,54–57], $\epsilon_\ell$ is assumed to be a single value applicable to the bulk of the polymer solution. However, inside the coil-like polymer conformations, $\epsilon_\ell$ cannot be treated as a single constant, and we expect inevitable variations in the value of local dielectric constant $\epsilon_\ell$.

One of the grand challenges in the physics of biomacromolecules and synthetic charged macromolecules is to achieve a fundamental understanding of the local dielectric function in the immediate neighborhood of ionic species in crowded conditions. To glean insight into this challenge, polyzwitterions with zwitterionic pendant groups (where one ionic group is closer to the chain backbone of lower dielectric constant, and the other ionic group is exposed to the bulk electrolyte solution of higher dielectric constant) offer an opportunity to explore any discriminatory behavior of the two ionic groups in terms of their different levels of counterion binding. In view of this, we have chosen polyzwitterions as the ideal case to explore the effect of variations in the local dielectric constant.

In general, in bulk dilute solutions, a dispersed flexible polyzwitterion adopts enormous number of conformations and there is a perpetual change in its conformations. Concomitant with these conformational fluctuations, the dipolar orientations of the zwitterionic repeat units are constantly changing. The net outcome is that these orientations and chain conformations are averaged out resulting in a coil-like statistical fractal object. Such a conformational status of the molecule prevents the resolution of the distinction between the local dielectric constants around the ionic groups on the repeat unit in terms of their normal distance from the chain backbone, even though the net effect of local dielectric heterogeneity must be present. This is also true with zeta potential measurements as well as the standard gel electrophoresis (with the typical large and heterogeneous mesh sizes), which measure the average properties. In the case of gel electrophoresis, if we could synthesize gels with nanoscopic mesh sizes, there is a chance of reaching our goal. The ultimate limit of such small meshes is the single nanopore system. Therefore, we have designed our experiment by trying to transport single polyzwitterion molecules through a nanopore as an unfolded single file. In this manner, the ionic groups constituting every repeat unit are exposed locally in contrast with the situation of heavily interpenetrating chain segments inside a coil.

With the above judicious choice of the polymer and the experimental technique, we chose silicon nitride solid-state nanopore (SSN) for the single-molecule electrophoresis instead of protein nanopores embedded in phospholipid membrane to avoid any complications from complexation between the polyzwitterion and the membrane. Even the SSNs in the present study can carry positive or negative charges (depending on pH) inside their lumen creating electro-osmotic flow (EOF)[18,41,47,58] that always accompanies the electrophoresis of the molecule. If the nanopore is positively charged, the EOF creates a fluid flow towards the positive electrode, and EOF is towards the negative electrode if the pore is negatively charged. Thus, depending on the sign of the pore charge and the nature of the macromolecule, EOF can either enhance or oppose the electrophoretic mobility of the macromolecule through the nanopore. In addition, charged pores can interact with the molecule affecting its electrophoretic movement. These effects must be accounted for in discerning the electrophoretic mobility of the macromolecule. Furthermore, since the neutral zwitterions are permanent dipoles, any effect from dielectrophoresis[59] arising from much weaker induced dipoles under nonuniform electric fields can safely be ignored. With the above described conceptual contexts and experimental design, we have investigated the ionic states of the neutral zwitterionc groups by accounting for all of the above mentioned contributing factors.

Here, using single-molecule electrophoresis, we demonstrate that the charge equivalence between the two ionized groups in the neutral zwitterionic monomers of neutral polyzwitterions is broken and the charge of the zwitterion's ionic group near the chain backbone is more shielded than that of the ionic group more distant from the chain backbone. We find that this unexpected phenomenon of charge symmetry breaking (CSB) is universal for all polyzwitterions independent of the zwitterion's dipole orientation to the chain backbone, and counterion identity. Considering the effects from EOF, pore-polymer interaction, strong electric fields at the interface, gradient in the local dielectric constant, and free energy barriers arising from polymer conformations under confinement, we show that the CSB effect arises from differential counterion binding to the ionic groups in the zwitterion due to gradients in the local dielectric constant. Further, we introduce a zeroth-order theoretical model allowing a quantitative determination of the extent of CSB from the voltage dependence of translocation time.

## Results

### Translocation experiments and charge symmetry breaking

The experimental setup (see "Methods") is a flow cell consisting of two chambers (donor and receiver) with a 10 nm-thick silicon nitride membrane in between. We have fabricated a nanopore of desired pore diameter (3.5–3.7 nm) within the membrane using the controlled dielectric breakdown method[60,61]. The choice of pore diameter was made to allow essentially single-file passage of extended conformations of poly(sulfobetaine methacrylate) (PSBMA) and poly(2-methacryloyloxyethyl phosphorylcholine) (PMPC) and to permit neither simultaneous occupancy of two chains inside the pore nor passage of the molecule as multiply folded blobs. Using the OVITO software[62] the cross-sectional diameter of these molecules is ~2.2 nm without accounting for hydration around the zwitterion moieties. Computer simulations[63] have shown that water molecules are bound to polyzwitterion backbone that can extend to multiple layers and that the mobility of water molecules near the backbone returns to the bulk solvent mobility only at about 1.8 nm from the backbone. Of course, most of the water molecules bound to the polymer are expected to be stripped away when the polymer undergoes translocation. However, as a conservative estimate, if we take the length of the pendant zwitterion group to be extended by an effective layer of even one strongly bound water (of length ~0.28 nm), the effective cross-sectional diameter of the polymer is ~2.8 nm. Therefore, the choice of pore diameter of 3.5–3.7 nm would prevent the simultaneous passage of more than one chain through the nanopore. The monomer length is estimated as 0.25 nm (using OVITO) and the average contour lengths of PSBMA and PMPC are estimated as 128 nm and 31.5 nm, respectively, which are longer than the pore length. After filling the flow cell with 1 M KCl, 10 mM HEPES, pH 7, the ionic current through the open pore is measured by applying an external electric potential difference $\Delta V$ across the two chambers of the flow cell. After adding 100 nM PSBMA (Fig. 1a) to the donor chamber, we have recorded the ionic current as a function of time.

The main result is shown in Fig. 1 for the single-molecule electrophoresis of PSBMA.

At the experimental pH 7, the polymer is neutral (because p$K_a$ of the sulfonic acid is around ~7 so that it is fully deprotonated and the quaternary ammonium group carries a permanent positive charge) as evident from its pH titration curve (Fig. 1b). In the positive configuration (Fig. 1c) of the chambers, where the positive electrode is in the donor chamber, the measured ionic current through the nanopore without PSBMA is 0.95 nA at the applied voltage of 150 mV. Upon addition of 100 nM PSBMA into the donor chamber, the measured ionic current remains the same as the open pore current without any transient blockages that would be the earmark of PSBMA translocation (marked as "no effect" in Fig. 1d). This result is consistent with the

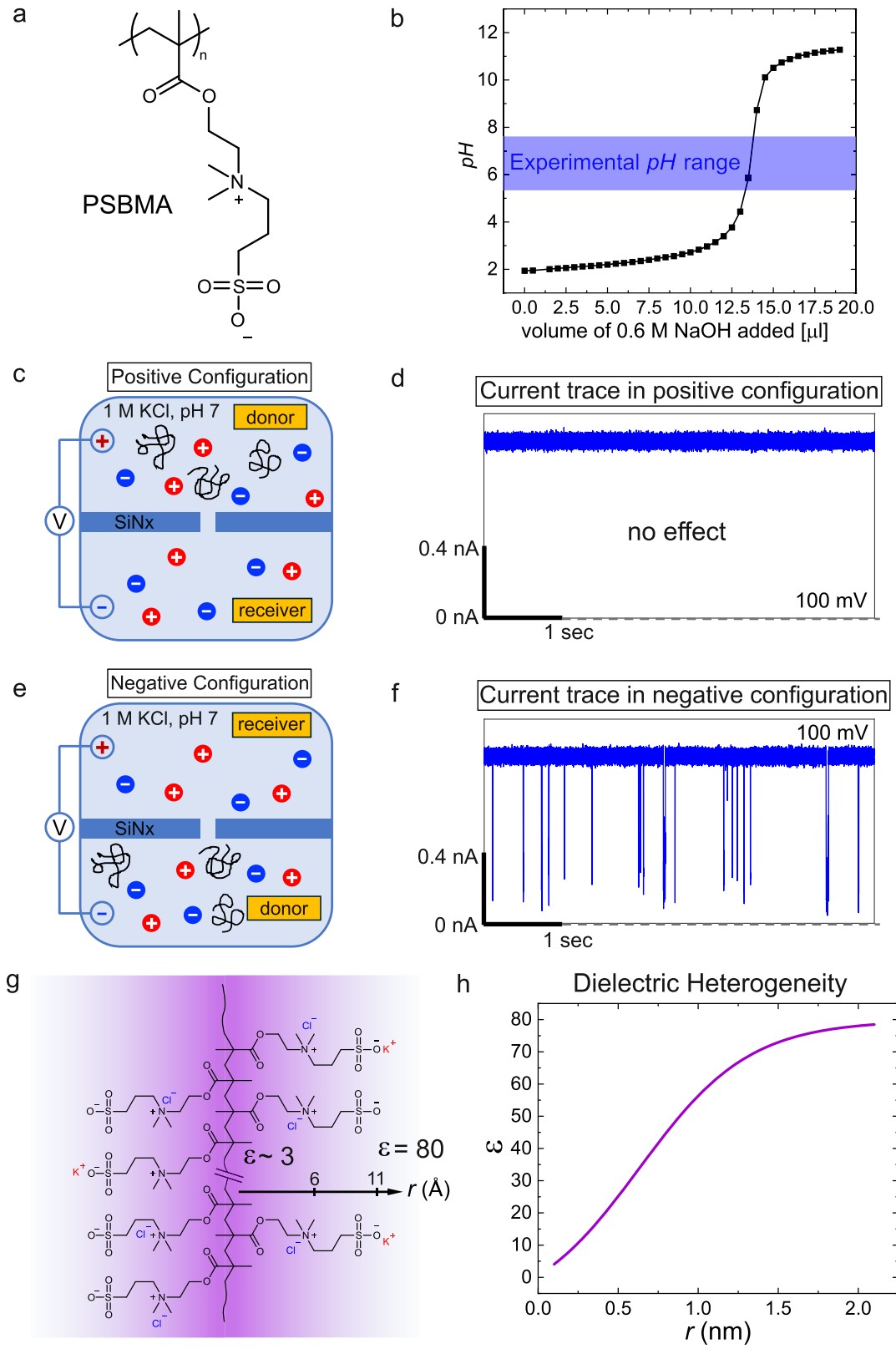

**Fig. 1 | Charge symmetry breaking in neutral PSBMA. a** Chemical formula of PSBMA with positively charged group closer to the chain backbone and the negatively charged group away from the chain backbone. **b** Titration curve of PSBMA (80 mM in 1 mL of 1 M KCl) showing charge neutrality at the experimental pH range. In the positive configuration of the experimental setup (**c**), PSBMA does not show any translocation event (**d**). In the negative configuration of the experimental setup (**e**), PSBMA exhibits a series of translocation events, behaving like a polyanion (**f**). **g** Differential binding of counterions to the positive and negative groups of PSBMA, originating from the gradient of local dielectric constant, causes CSB. **h** Sketch of the local dielectric constant as a function of the distance away from the chain backbone. In (**c**–**f**), solution: 10 nM PSBMA in 1 M KCl buffered at pH 7 with 10 mM HEPES, pore diameter: 3.5 nm, voltage: 100 mV. In (**d**–**f**), sampling rate: 250 kHz, low-pass filter frequency: 10 kHz.

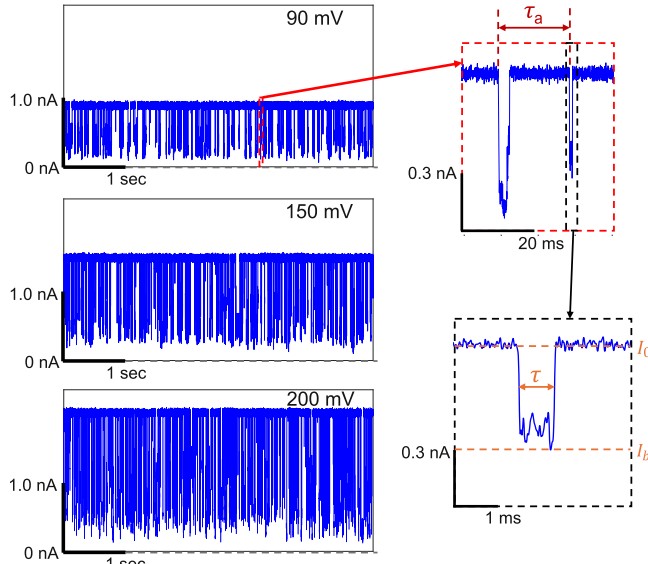

**Fig. 2 | Voltage dependence of ionic current blockages for PSBMA translocation in negative flow cell configuration and their quantification.** 100 nM PSBMA was used in 1 M KCl buffered at pH 7 with 10 mM HEPES. Pore diameter: 3.7 nm, sampling rate: 250 kHz, low-pass filter frequency: 100 kHz. Gaussian filter frequency: 10 kHz (Left) Dependence of the ionic current trace, showing a series of temporary current blockages, on voltage (90 mV, 150 mV, and 200 mV, from top to bottom). (Right) Expanded view of selected current blockage events. $\tau_a$ is the inter-arrival time between two successive blockage events; $I_0$ is the open pore current, $I_b$ is the blockage current corresponding to the minimum of the blocked current, and $\tau$ is the full-width at half maximum, identifying the blockage duration.

thought that neutral polymers hardly can undergo electrophoretic translocation.

However, remarkably, in the negative configuration (Fig. 1e), where the negative electrode is in the donor chamber, upon addition of 100 nM PSBMA in the donor chamber, the molecules start translocating through the nanopore one at a time as evident from the ionic current blockages in the ionic current trace (Fig. 1f). As described below, most of these blockages are successful translocation events. Thus, PSBMA behaves like a polyanion even though it is neutral at the experimental pH (Fig. 1b). This means that the charge symmetry between positive and negative charges−where one positive and one negative charge balance each other to maintain neutrality (charge symmetry)−constituting the zwitterionic group is broken. We show below that this unexpected phenomenon of CSB is universal for all polyzwitterions independent of the zwitterion's dipole orientation to the chain backbone, and counterion identity.

We attribute CSB to the dielectric heterogeneity around polymer backbones. As already alluded to, since $\Delta G$ is inversely related to $\epsilon_\ell$, the counterion binding is stronger in regions with lower dielectric constant. As expected for polymers in aqueous solutions, the local dielectric constant is low near the oil-like chain backbone (Fig. 1g), and continuously increasing to ~80 in the bulk solution away from the chain (sketched in Fig. 1h). Therefore, the ionic group of the zwitterion closer to the backbone is subjected to more counterion binding (and thus more neutralized) compared to the ionic group further away from the chain backbone. This differential counterion binding makes the charge of the outer ionic group dominant (CSB).

The cause of the observed polyanionic behavior of PSBMA might, in principle, arise from several factors that include electrophoretic mobility due to differential counterion binding, EOF, stronger electric fields near the charged pore wall affecting peripheral ionic group more selectively, and stronger electric fields at the pore entrance. After careful scrutiny of these various potential causes given below, we

conclude that the above-described mechanism of differential counterion binding across a gradient in the local dielectric constant is responsible for CSB.

The isoelectric point of our silicon nitride pore is about pH 6 (Supplementary Fig. 1, consistent with literature value[64]), and the charge density outside pH 6 is only weak[65]. In the experimental condition of pH 7 for PSBMA, the pore lumen is negatively charged so that the EOF is towards the negative electrode[10,47]. Since PSBMA moves towards the positive electrode, the EOF is overwhelmed by the electrophoretic force resulting in the polyanionic behavior of PSBMA. Furthermore, the calculated radial electric field from the pore wall decays sharply within a very short distance comparable to the short Debye length corresponding to 1M KCl solution in the experiment[47]. Since such electric field gradients do not depend on the externally applied voltage across the pore, the radial field gradient cannot be attributed as the origin for the observed voltage dependence of rectified mobility of the polyzwitterion. Also, the strength of the electric field along the field direction is uniform inside the nanopore (where translocation occurs) except at a very thin skin at the pore wall[47]. Furthermore, stronger electric fields at the pore mouth result in enhanced localization of dipolar polymers at the pore mouths which increases the free energy barrier for the subsequent translocation. This will result in the opposite consequence on the voltage dependence of the translocation time compared to our experimental observation. Furthermore, such localized capture at the pore mouth, on its own, cannot lead to the observed voltage polarity dependence in the current blockades. Therefore, CSB cannot be attributed to any inhomogeneity of the electric field or EOF.

As a further validation of CSB, the frequency of arrival of molecules inside the pore entrance (capture rate $R_c$) increases with voltage (Fig. 2, left) in the negative flow cell configuration, consistent with PSBMA behaving like polyanions. Following previous works on single-molecule electrophoresis[11,14,38,42], $R_c$ is obtained as the inverse of the average time duration $\tau_a$ (Fig. 2, top right) between the starting times of two consecutive blockage events (see "Methods"). Furthermore, to relate the magnitude of CSB with the effective charge of the dipoles and local dielectric constant, we have measured the blockage current ratio $I_b/I_0$ (where $I_0$ is the open pore current and $I_b$ is the minimum current in the blockade) and the blockage duration ($\tau$) as the full width at half maximum of the current blockage (Fig. 2, bottom right). $\tau$ is translocation time when the blockages correspond to successful translocation events (see below).

The dependence of $R_c$ on the voltage is given in Fig. 3a, showing a linear dependence of $R_c$ on $\Delta V$. In general, $R_c$ is controlled by thermal diffusion, electrophoretic drift, and free energy barrier at the pore[26,27]. As a result, $R_c$ is essentially zero at very low $\Delta V$, then increases nonlinearly with $\Delta V$ in the barrier-dominated regime, and finally is directly proportional to $\Delta V$ in the drift-dominated regime. This general trend of $R_c$ is seen for PSBMA. At voltages higher than 70 mV, $R_c$ depends linearly on the applied voltage (Fig. 3a, error bars are from 95% confidence intervals) corresponding to the drift-dominated regime. By extrapolating the linear line to lower voltages until the intersection with zero $R_c$, the threshold voltage for the onset of the drift-dominated regime is obtained as 29 mV, below which $R_c$ is essentially zero corresponding to the barrier dominated regime. The threshold value implies a free energy barrier for PSBMA capture, which arises from a combination of the entropic barrier for one chain end of jammed polymer coil to be inserted into the pore and polymer conformational entropy.

In the drift-dominated regime, the slope of capture rate versus voltage ($4.7 \times 10^{-3}$ Hz nM$^{-1}$ mV$^{-1}$ in Fig. 3a) is proportional to $c\mu M^{-1}$, where $c$ is the concentration of PSBMA, $\mu$ is the electrophoretic mobility, and $M$ is the pore length. This constant slope indicates that the effective charge of PSBMA chains is a constant and negative in the donor chamber.

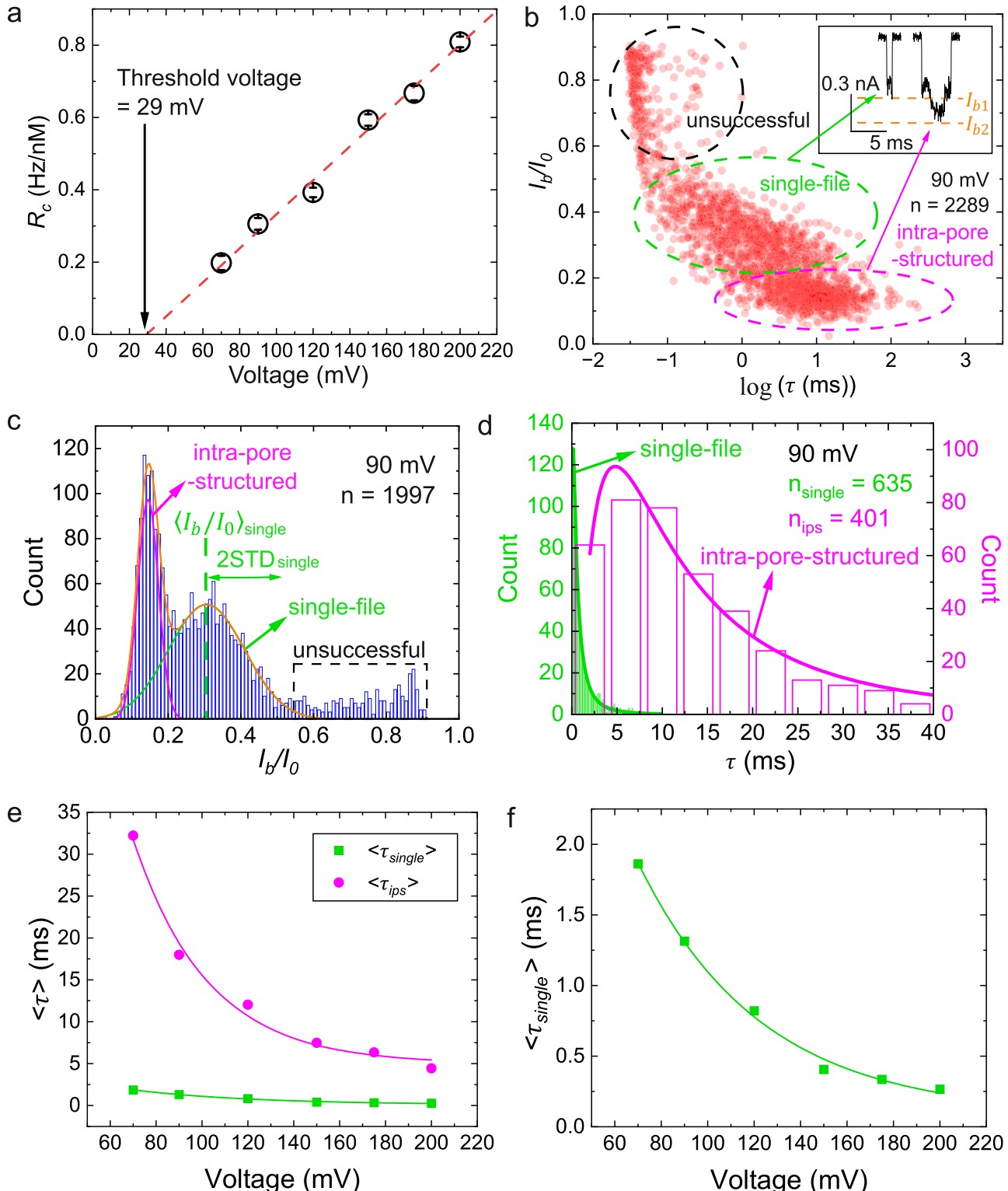

**Fig. 3 | Capture rate and translocation kinetics for PSBMA.** 100 nM PSBMA in 1 M KCl buffered at pH 7 with 10 mM HEPES was used. Pore diameter: 3.7 nm, sampling rate: 250 kHz, low-pass filter frequency: 100 kHz, Gaussian filter frequency: 10 kHz. **a** Linear dependence of capture rate $R_c$ on voltage showing a threshold for capture of PSBMA. Error bars are from 95% confidence intervals. **b** Three populations in the event scatter plot of blockage current ratio $I_b/I_0$ versus logarithm of blockage duration $\tau$, identified as unsuccessful, single-file chain, and intra-pore-structured chain translocations (90 mV). Details of blockages for single-file and intra-pore-structured translocations in the inset. **c** Histogram of $I_b/I_0$ and deconvolution into unsuccessful, single-file, and intra-pore-structured translocation events (90 mV). Curves are Gaussian fittings. $n$ is the number of events used for double Gaussian fitting. **d** Histograms of the translocation time for the single-file and intra-pore-structured translocations (90 mV). Curves are log-normal distribution functions. **e** Dependence of mean translocation time $\langle\tau\rangle$ on voltage (magenta circle: intra-pore-structured; green square: single-file). $n_{ips}$ and $n_{single}$ are the numbers of events used for log-normal fitting of intra-pore-structured translocations and single-file translocations, respectively. **f** Expanded view of voltage dependence of $\langle\tau\rangle$ for the single-file chain translocation. Curves in (**e**, **f**) are fittings to exponentials.

The kinetics of translocation is as follows. The earmark of successful translocation of a molecule through a nanopore is the inverse dependence of the average translocation time ($\langle \tau \rangle$) on the applied voltage $\Delta V$[10,13,28,32,45]. $\tau$ is highly stochastic and broadly distributed (Fig. 3b). Conjugate to the distribution of $\tau$, $I_b/I_0$ is also broadly distributed. The mutual distributions of these quantities for PSBMA at 90 mV is displayed in the event scatter plot (Fig. 3b) of a total of 2289 events. Please see Supplementary Fig. 2 for scatter plots at other voltages. The data in Fig. 3b fall into three populations marked by ellipses. The first population at the top of the event scatter plot (black-dashed ellipse) consists of shallow blockages of very short duration corresponding to collisions of the PSBMA molecules with nanopore entrance, as in the previous studies of polyelectrolyte translocation[11]. These are labeled as unsuccessful events (Fig. 3b).

The second population (green-dashed ellipse) in the middle has moderate blockages, while the third population (magenta-dashed ellipse) at the bottom has deep and long blockages. Both of these populations correspond to translocation processes, because their average translocation times decrease with voltage (see below). Most of the moderate blockages have one blockage level as shown in Supplementary Fig. 3. The $I_b/I_0$ histograms fitted with multiple Gaussian distributions show that the mean $I_b/I_0$ for the moderate events is -0.4, as in the example at 90 mV given in Fig. 3c. Please see Supplementary Fig. 4 for other voltages and Methods for fitting details. This mean blockage current ratio corresponds to the cross-sectional diameter of polymer as ~2.9 nm calculated by $I_b/I_0 = 1 - (a/d)^2$, where $a$ is diameter of polymer and $d$ is pore diameter (3.7 nm). Based on the effective cross-sectional diameter of the hydrated polymer ( ~2.8 nm), we assign the second population as the single-file translocation events. Even in this single-file mode, we expect weak conformational fluctuations due to chain flexibility and rearrangement of bound water, etc. as sketched in Supplementary Fig. 5. On the other hand, deep blockade events show complex features in the ionic current as in Supplementary Fig. 6. At least 93% of these events show two different blockage levels within one event as in the inset of Fig. 3b with one blockage level comparable to the blockage level of the moderate events. Although these two levels are not sharply quantized in contrast with dsDNA translocations of folded chains[16,29,44], these are translocation events, because of decrease of their average duration with voltage, which is shown later.

We conjecture that the population of events with deeper blockades corresponds to translocation events where the polymer adopts conformations with one or more local short hernia-like structures inside the nanopore, as cartooned in Supplementary Fig. 7. Such structures can easily arise in the present system due to the flexibility of the chain as well as local association of dipolar monomers. We call the mode of translocation of intra-pore conformations with local kinks or nano-cilia (or nano-blobs) as "intra-pore-structured (ips)" translocation. See Supplementary Note 1 for detailed discussions.

The proportion of the single-file translocation decreases with voltage (see Supplementary Fig. 8) indicating that higher voltages enhance intra-pore-chain structures during translocation. Furthermore, it is possible for the chain to additionally form doubly kinked conformations during translocation at higher voltages (175 mV and 200 mV) with even deeper and longer blockages (see Supplementary Fig. 9). During these translocations, most of the current blockages exhibit three different levels. In general, it is desirable to make an one-to-one correspondence between the polymer conformation inside the nanopore and the ionic current. This is a difficult task even for strong polyelectrolytes such as polystyrene sulfonate and DNA which have high charge density. The difficulty is more severe for flexible chains with very weak charge density that are investigated in the present study. Nevertheless, despite the impossibility to precisely identify the conformational details behind the very short-time features of the various ionic current traces, the large-scale behavior of translocation and the consequent CSB are evident from the data.

To separate and avoid the overlap of single-file and ips translocations in constructing $\tau$ histograms for each population, we have taken only the right half of the Gaussian distribution for the single-file translocation events and the left half of that for the ips translocation events. The histograms of the translocation time ($\tau$) for the separated single-file translocation events and the ips translocation events at 90 mV are given in Fig. 3d as an example (for other voltages see Supplementary Figs. 10 and 11). While histograms for the single-file translocations are narrow with shorter translocation times, histograms for ips translocations at the same voltages are broader with very long translocation times. The longer translocation time for the ips translocation events is presumably due to enhanced friction of PSBMA against the pore. The shapes of the histograms are in conformity with that of generic drift-diffusion process[10,12,13]. Following the previous practice in the literature[37,45], we have fitted these $\tau$ histograms with log-normal distributions in the range of $0 < \tau < 10$ ms, and $0 < \tau < 100$ ms for the single-file, and ips translocations, respectively. The average translocation times ($\langle \tau \rangle$) for these two processes are obtained from the log-normal fittings, and their dependence on the voltage for the single-file and ips translocations is portrayed in Fig. 3e f.

For both kinds of translocation, $\langle \tau \rangle$ decreases roughly exponentially with voltage, implying that overcoming a free energy barrier arising from the loss of conformational entropy during translocation is a dominant factor in determining the voltage dependence of $\langle \tau \rangle$, analogous to the behavior of single-stranded DNA translocation through solid-sate nanopores[30]. As mentioned above, this decreasing dependence of $\langle \tau \rangle$ with voltage is taken as the signature of successful translocation.

## Universality of CSB in polyzwitterions

Since we attributed CSB in PSBMA to stronger counterion binding at the positively charged proximal ionic group due to lower local dielectric constant near the chain backbone, compared to the negatively charged distal ionic group, one strategy to validate the generality of CSB is to change the identity of counterion (LiCl instead of KCl) in the above PSBMA tranlocation experiments. Furthermore, CSB should still be present when the the dipolar orientation of the zwitterionic group is reversed, where the negatively charged group is proximal and positively charged group is distal. In this case, the polyzwitterion should move towards the negative electrode and there should not be any translocation towards the positive electrode.

In view of these strategies to validate the mechanism of CSB and its general occurrence, we have investigated the single-molecule electrophoresis of (1) PSBMA in 3.6 M LiCl, 10 mM HEPES, pH 6 and (2) PMPC, with proximal negatively charged phosphoryl group and distal positively charged choline group constituting the zwitterion group attached to the backbone, in 3.6 M LiCl, 10 mM HEPES, pH 3.7. Note that at pH 3.7, the pore surface is positively charged leading to EOF towards to the positive electrode with the MPC moiety being net charge neutral in the absense of CSB. As shown below, these two systems exhibit the predicted CSB demonstrating the universality of the CSB in polyzwitterions.

The independence of CSB on salt identity is found as follows. By changing the salt identity to 3.6 M LiCl (pH 6), instead of 1 M KCl (pH 7) in Figs. 1c–f, 2, and 3, examples of ionic current traces for PSBMA are in Fig. 4a, b at the illustrative voltage of 200 mV. Again, we did not find current blockages in the positive configuration (Fig. 4a), but find blockage events in the negative configuration (Fig. 4b). Following the same procedure of data analysis used above for PSBMA in 1 M KCl (see Supplementary Figs. 12, 13, 14, and 15 for event scatter plots, $I_b/I_0$ histograms, single-file translocation time histograms, and ips translocation time histograms, respectively), $R_c$ depends linearly on $\Delta V$ indicating drift-dominated capture (Fig. 4c). The slope of the line in this regime is weaker (by a factor of about 3) than in the case of 1 M KCl, primarily due to weaker net charge arising from weaker extent of CSB

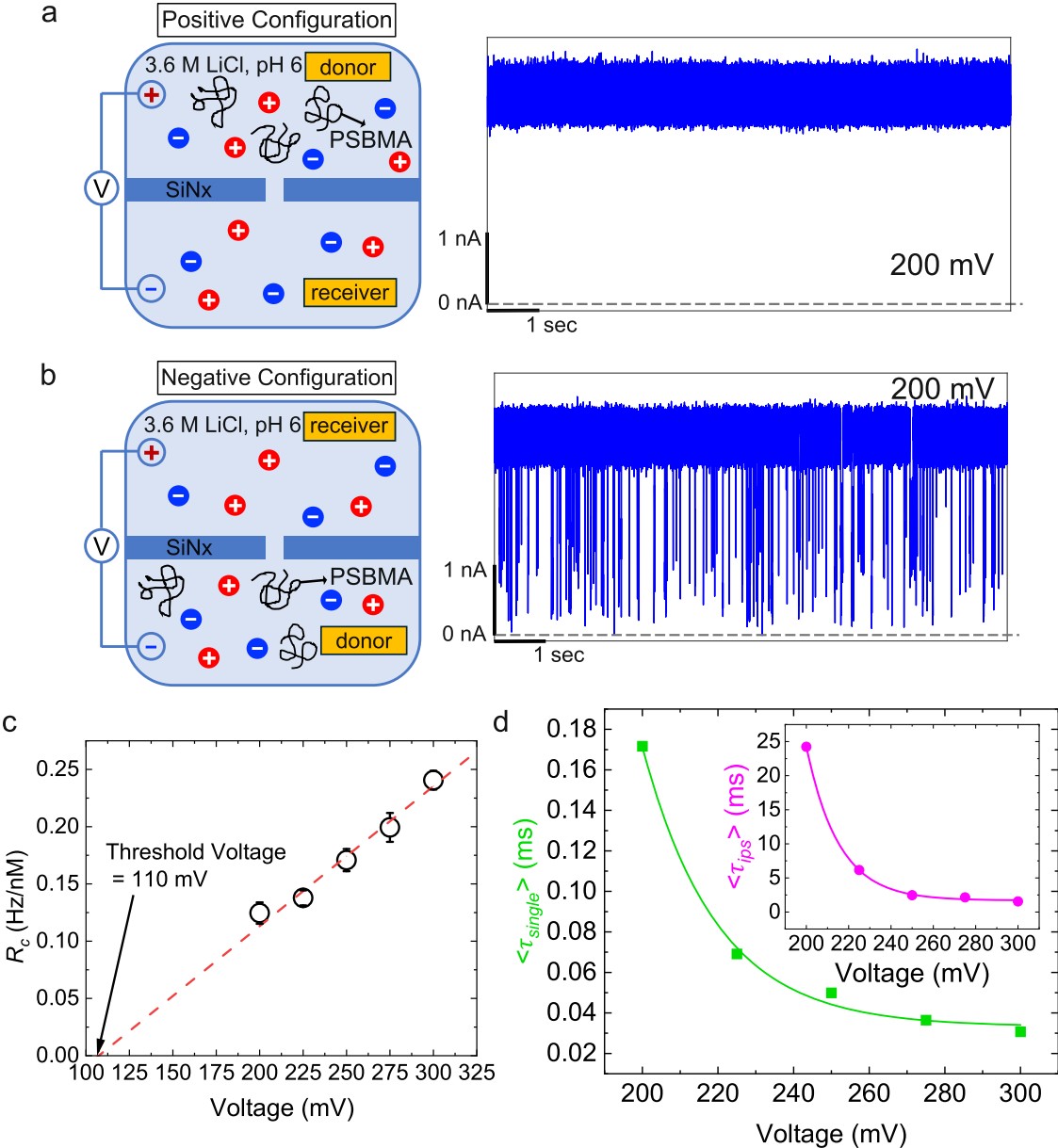

**Fig. 4 | Universality of CSB in neutral PSBMA independent of identity of counterion.** 100 nM PSBMA in 3.6 M LiCl buffered at pH 6 with 10 mM HEPES was used. Pore diameter: 3.7 nm, sampling rate: 250 kHz, low-pass filter frequency: 100 kHz. **a** No translocation towards negative electrode in 3.6 M LiCl. **b** Occurrence of translocation towards positive electrode, whereby PSBMA acts like a polyanion as in Figs. 1 and 3. **c** Linear voltage dependence of capture rate $R_c$ with a threshold value of voltage (110 mV). Error bars are from 95% confidence intervals. **d** Voltage dependencies of mean translocation time $\langle \tau \rangle$ for single-file (main) and intra-pore-structured (inset) translocation events. Curves are fittings to exponentials.

and higher viscosity of the solution. Extrapolation of this linear relation to lower voltages yields a threshold voltage of 110 mV that is representative of the free energy barrier that needs to be crossed for successful capture. The higher value of the threshold voltage for 3.6 M LiCl, compared to the lower value for 1 M KCl, is due to a lower net charge after breaking the charge symmetry.

As in the case of 1 M KCl solution, we find translocation events corresponding to both single-file single-file chain conformations and ips conformations of PSBMA. Using the same data analysis protocol to identify the single-file and ips translocations, the voltage dependencies of $\langle \tau \rangle$ for the single-file (main curve) and ips translocations (inset) are given in Fig. 4d. In both cases, $\langle \tau \rangle$ decreases roughly exponentially with the voltage confirming again that PSBMA undergoes successful translocation like a polyanion even with the different salt identity.

The independence of CSB on the dipole orientation of polyzwitterions is demonstrated as follows. After confirming that at the experimental pH, PMPC (with dipole orientation opposite to PSBMA) (Fig. 5a) is charge neutral as shown in its pH titration curve (Fig. 5b) due to low pK$_a$ value of phosphoric acid group (≤1), we recorded ionic current traces in the positive and negative flow cell configurations with a pore diameter of 2.8 nm and 100 nM PMPC. In the negative configuration, where the negative electrode is in the donor chamber containing PMPC, there are no translocation events as shown in the example at 300 mV in Fig. 5c. However, current blockage events occur in the positive configuration (Fig. 5d) at the illustrative voltage of 300 mV. Here, PMPC molecules move toward the negative electrode acting like a polycation (in contrast with PSBMA acting like a polyanion) thus validating the general premise of CSB. Since the direction of electrophoretic mobility of PMPC is in the opposite direction to

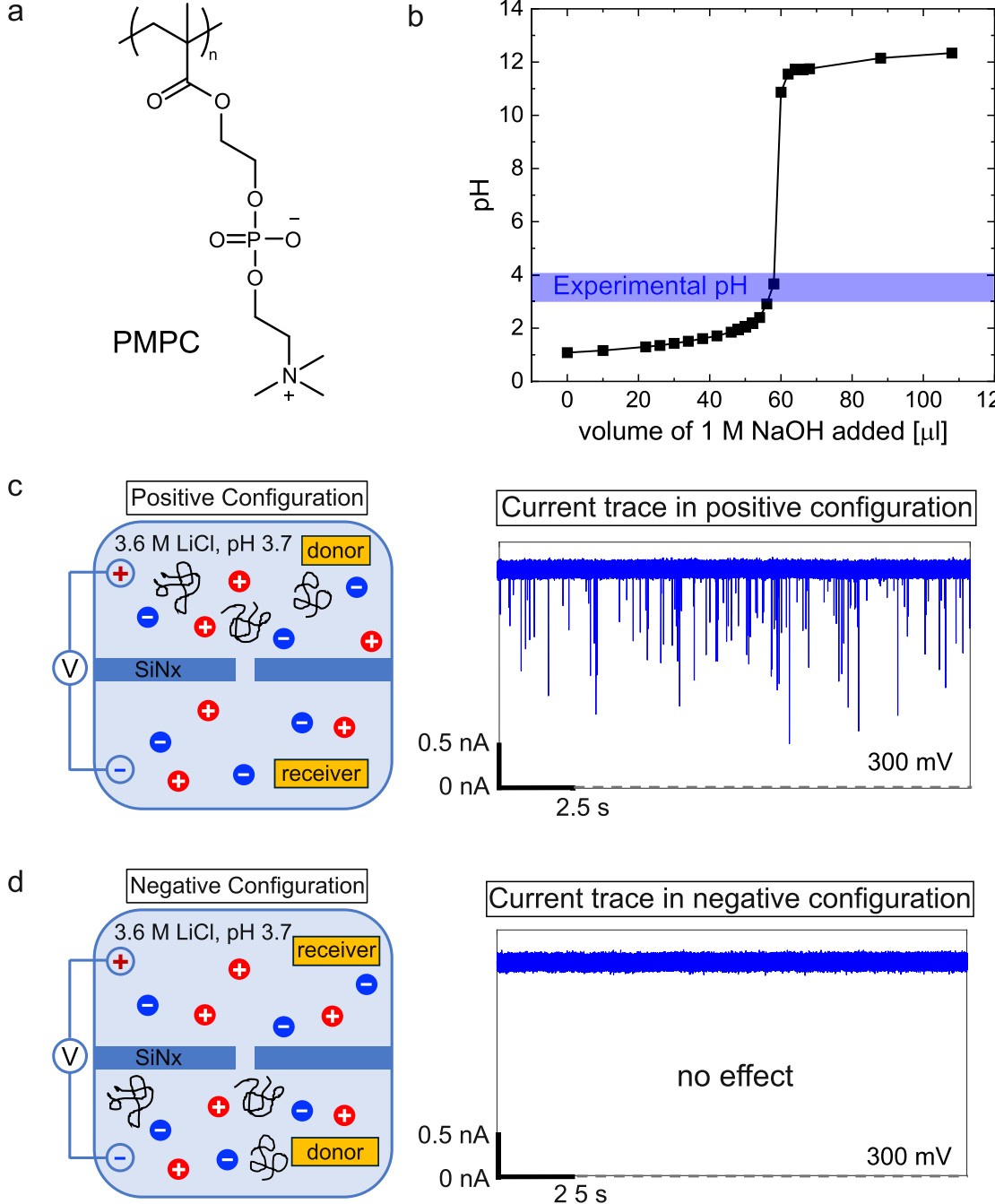

**Fig. 5 | Charge symmetry breaking in PMPC. a** Chemical formula of PMPC. **b** Titration curve of PMPC (63 mM in 1 mL of Milli Q water) showing that PMPC is neutral at the experimental pH 7. **c** PMPC translocates to the negative electrode acting like a polycation. **d** PMPC does not translocate to the positive electrode. In

(**c**) and (**d**), 50 nM PMPC in 3.6 M LiCl buffered at pH 3.7 with 10 mM HEPES was used. Pore diameter: 2.8 nm, voltage: 300 mV, sampling rate: 250 kHz, low-pass filter frequency: 10 kHz.

EOF, the role of EOF is overwhelmed by the electrophoretic force, as in the case of PSBMA. Further in-depth analysis of translocation kinetics of PMPC and other polyzwitterions is of high interest in our future work.

Combination of the above results on PSBMA with different salt identities and PMPC comprehensively vouches for the universal occurrence of CSB in polyzwitterions.

**Theory of effective charge and translocation time**
In addition to exhibiting CSB, the above experiments provide quantitative values of $\langle \tau \rangle$ which must be a function of the individual effective

charges (due to differential counterion binding) of the ionic groups of the polyzwitterion, which in turn are due to gradients in local dielectric constant. To deduce the charges from $\langle \tau \rangle$ and to estimate the local dielectric constant, we use theory. Understandably, the task of formulating a rigorous theory for these systems is difficult due to the complexities arising from a confluence of electrostatic and hydrophobic interactions, conformational entropy of polymer chains, solvent reorganization, and liquid crystal-like dipolar orientational correlations[53]. Nevertheless, to explore the root cause of CSB, we present the following zeroth-order mean field theory and establish a relation between the macroscopically observed CSB and the

microscopic nature of differential counterion binding within zwitter-ionic monomers of the polymer. The cross-sectional diameter that can be qualitatively inferred from the ratio of blocked current in single-file blockages to the open pore current is -2.9 nm, consistent with the value we obtained from molecular mechanics. Further, the radius of gyration $R_g$ (estimated as -9.2 nm) is more than twice bigger than the pore diameter. Therefore, there is no possibility for the chain to translocate as either as a single blob or a sequence of blobs which would be unfavored due to conformational entropic penalty. In view of these considerations, we identified the mode of translocation as threading of extended conformations as in the well-established nanopore-based sequencing of polynucleotides and proteins.

There are four key elements in the theory. (1) We present an expression for the effective degree of ionization $\alpha$ (equivalently the effective charge $q$) of the ionic groups of the zwitterion in terms of the local dielectric constant $\epsilon_\ell$, based on counterion-binding equilibria (see Supplementary Note 2) as

$$\alpha = \left(1 + [A^-]\exp\left(\frac{e^2}{4\pi\bar{\epsilon}\epsilon_\ell k_B T r}\right)\right)^{-1}, \tag{1}$$

where $[A^-]$ is the activity of the counterion.

(2) We present a zeroth-order model for the translocation kinetics, capable of capturing the essential physical concepts that contribute to the phenomenon, and yet mathematically simple enough to derive useful formulas to compare with experiments. The chemical details of the polyzwitterion molecule are many that include the size and orientation of the zwitterionic repeat unit with respect to the chain backbone, distance between the ionic groups and dipole moment of the zwitterionic unit, separation distance between two adjacent repeat units, thickness of the backbone, chain length of the polymer, pore length, pore diameter, surface charge density on the inner wall of the pore, ionic strength, pH, and identity of counterions, and physical quantities such as local dielectric constant and externally imposed electric field profile across the nanopore. Inevitably, these contributing factors require parametrization which is not arbitrary. As described in Supplementary Note 3, we have taken reasonable values of these parameters for the specific polyzwitterion systems studied here in setting up the model for the next step of the calculation.

(3) Using the model, we present the free energy landscape for the translocation of the molecule as sketched in Fig. 6 and described in Supplementary Notes 4 and 5. Briefly, the translocation occurs in three stages: crossing an entropic barrier to enter the pore, threading through the pore, and successful ejection from the pore. The free energy landscape $F(m)$ arising from these three stages is a composite of four contributions: charge-electric field interaction ($F_{pE}$), conformational entropy of the chain ($F_{ent}$), pore wall-chain interaction ($F_{pore}$), and electrostatic energy ($F_{bE}$) of the chain in the bulk of the receiver chamber. The net result for the free energy is

$$F(m) = F_{pE} + F_{ent} + F_{pore} + F_{bE}, \tag{2}$$

where $m$ denotes the translocation coordinate in units of monomer length. These contributions are presented in Supplementary Note 4 and summarized in Supplementary Tables (I and II).

(4) Based on the free energy landscape, using the Fokker-Planck formalism[66] (Supplementary Note 6), we calculate the average translocation time as a function of the voltage in terms of the unassigned values of the charges of the ionic groups, given by

$$\langle\tau\rangle = \frac{1}{k_0}\int_0^{N+M} dy\, e^{F(y)/k_B T}\int_0^y dz\, e^{-F(z)/k_B T}, \tag{3}$$

where $F(y)$ is given in equation (2), N and M are the contour length of the polymer and pore length in units of monomer length (0.25 nm),

and $k_0$ is the bare monomer diffusion coefficient inside the pore[10]. The important outcome of the theory is a fundamental understanding of the relative contributions from conformational entropic barrier, pore-polymer interaction, and electrophoretic drift to the translocation time. The calculated results are then compared with experimental results on the voltage dependence of the average translocation time, to determine the extent of CSB.

## Comparison between theory and experiment to quantify CSB

The parameters entering the theoretical prediction of $\langle\tau\rangle$ are the charge $q_1$ of the ionic group close to chain backbone, the charge $q_2$ of the ionic group away from the chain backbone, the local dielectric constant $\epsilon_\ell$, and $\Delta V$, in addition to the pore length and chain length which are fixed. For polyelectrolytes in aqueous electrolyte solutions, it is known from a combination of Manning theory[54,55], counterion adsorption theory[57], and experiments[17,56], that the effective degree of ionization is around -0.25. Therefore, we have taken $q_2 = -0.25$ as reported in the literature with local dielectric constant around 80. We have estimated the pore wall-chain interaction energy per monomer $\epsilon_0$ as $0.0009 k_B T$ (see Supplementary Note 7). As given in equation (3), the average translocation time $\langle\tau\rangle$ appears as $k_0\langle\tau\rangle$, where the monomer diffusion coefficient $k_0$ inside the pore is unknown. In view of this, to eliminate $k_0$, we have constructed the ratio of the calculated $k_0\langle\tau\rangle$ at a given voltage to that at a reference voltage (taken as 200 mV in Fig. 7a). $k_0$ is then determined by comparing the theoretical predictions and experimental data on $\langle\tau\rangle$ (see below). The theoretically calculated value of $\langle\tau\rangle/\langle\tau\rangle_{200mV}$ is given in Fig. 7a as a function of the voltage for different values of $q_1$. To deduce the value of $q_1$ and hence the magnitude of CSB, these curves are then compared with experimental data on $\langle\tau\rangle/\langle\tau\rangle_{200mV}$ for different voltages (in the range of 70–200 mV) (black data points, Fig. 7a). By minimizing the ratio $((\langle\tau\rangle/\langle\tau\rangle_{200mV})_{theory} - (\langle\tau\rangle/\langle\tau\rangle_{200mV})_{experiment})/(\langle\tau\rangle/\langle\tau\rangle_{200mV})_{theory}$ and its standard deviations (see "Methods"), we find the best value of the charge of the quaternary ammonium group as +0.10 using data of PSBMA translocation in 1 M KCl, 10 mM HEPES, pH 7 (Fig. 3f).

For the LiCl data, both the experimental and theoretical values of $\langle\tau\rangle/\langle\tau\rangle_{300mV}$ exhibit the same common trend of $\langle\tau\rangle/\langle\tau\rangle_{300mV}$ decreasing with voltage, as shown in Supplementary Fig. 16. However, experimental values of $\langle\tau\rangle/\langle\tau\rangle_{300mV}$ decrease more rapidly with voltage compared to the theoretical values. Perfect quantitative fitting between the present theory and experiments cannot be expected due to the well-recognized 'special ion' effect of Li$^+$ arising from its high electron density[67]. The Fokker-Planck formalism used in the theory is not capable of describing the specific details of hydrated Li$^+$ ion at different voltages. A more fundamental theoretical treatment of specificity of counterions in nano-confinement is beyond the scope of the present goal and is relegated to future investigations.

We note that the theoretically calculated translocation time is in units of the monomer friction coefficient $1/k_0$. To estimate $k_0$, we have taken $k_0$ as an unknown constant independent of voltage and found its value as $1.41 \times 10^4$ nm$^2$ ms$^{-1}$ from the best fit between the experimental values of $\langle\tau\rangle$ (black squares) and theoretical values (red circles) (Fig. 7b). The procedure for finding the best fit is the same as for obtaining $q_1$, by minimizing the mean and STD of the error between theory and experiment. With this value of $k_0$, the experimental data (black squares) and theoretical values (red circles) for the voltage dependence of $\langle\tau\rangle$ are given in Fig. 7b. The nice overlap between these two curves using a single value of $k_0$ indicates that all monomers of the chain in each translocation event have uniform velocity, without any additional nonlinear effects such as tension propagation forces seen in other systems[40].

The inference of local dielectric constant from CSB is performed as follows. The effective charge $q_1$ of the ionic group near the chain backbone, obtained using theory and experiments, is related to the

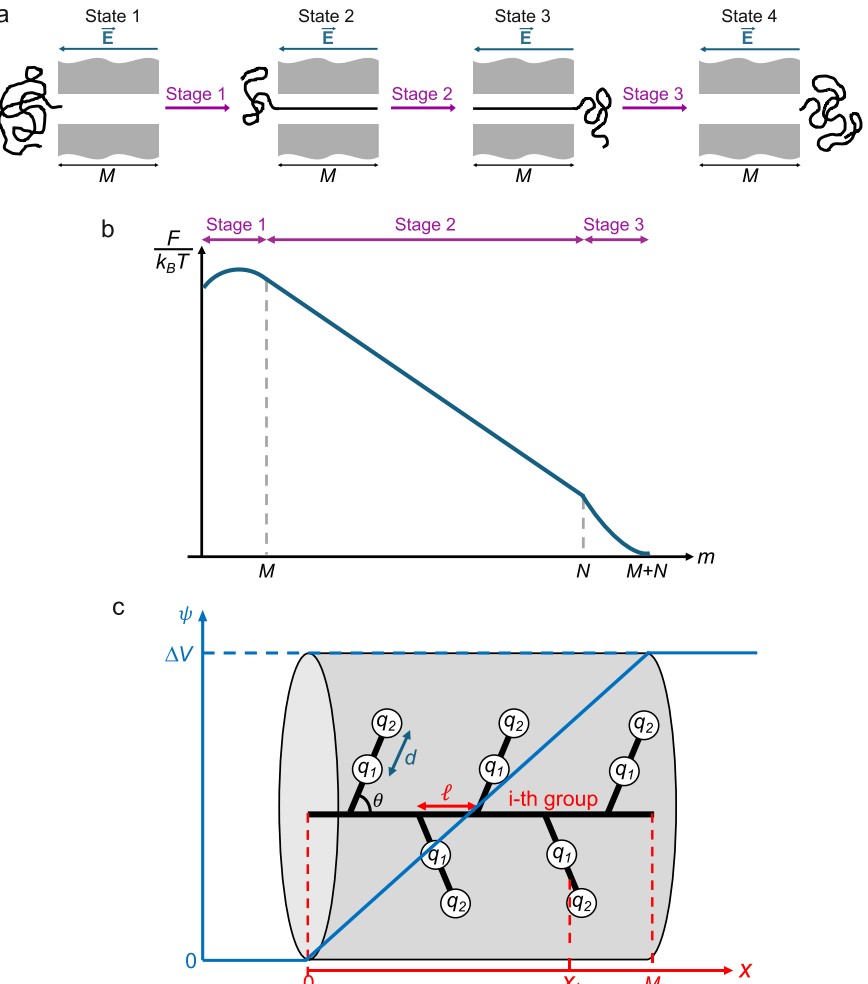

**Fig. 6 | Model of polyzwitterion translocation. a** Three stages of translocation. **b** Sketch of free energy ($F$) landscape corresponding to the three stages. $M$ and $N$ are the pore length and chain length in units of monomer length $\ell$. **c** Model of the conformation of a segment of a polyzwitterion chain inside the nanopore with its central axis along the $x$-coordinate in units of $\ell$. Blue curve denotes the profile of the electric potential $\psi$ due to the applied voltage. $q_1$ and $q_2$ denote the charges of the ionic groups closer and distant to the chain backbone, respectively. $d$ is the dipole length in units of $\ell$, and $\theta$ is the angle subtended by the zwitterionic group with respect to the chain backbone taken along the central axis of the nanopore. $x_i$ is the $x$-coordinate of the center of the $i$-th zwitterionic group.

local dielectric constant $\epsilon_\ell$ through equation (3), where $\alpha = q_1$. Assuming that the activity $[A^{-1}]$ of the counterion can be adequately approximated by its mole fraction, equation (3) yields a relation between $q_1$ and $\epsilon_\ell$. Choosing $r$ in the range of 0.3–0.5 nm, the local dielectric constant $\epsilon_\ell$ is in the range of 20–30. This estimate is consistent with the counterion binding energy being comparable to $k_B T$, by equating the counterion binding energies for the internal ionic group, $q_1 e^2/(4\pi\epsilon\epsilon_\ell r)$ and for the outer ionic group, $q_2 e^2/(4\pi\epsilon\epsilon r)$, with $k_B T$. Since we have taken $q_2 = -0.25$ from literature[17,55] and $q_1 = +0.1$ from our experiments, the local dielectric constant is 28, when the same value of $r$ is assumed for both groups and $\epsilon = 70$[67] for 1 M KCl solutions. Thus, generally, the local dielectric constant is lower than the value in the bulk solution. A more accurate deduction of $\epsilon_\ell$ is relegated to future work which might demand quantum-mechanical calculations. Nevertheless, the key quantity in CSB is its magnitude, that is the difference in the magnitude of the effective charges of the ionic groups of the zwitterion, instead of the absolute values of these effective charges.

## Discussion

We have addressed the fundamental question of whether macromolecules made of polar, but electrically neutral, monomers can move under an electric field and if so in what direction with what magnitude. Combining single-molecule electrophoresis experiments and theory, we have addressed this question by investigating two polyzwitterions (PSBMA and PMPC) that are electrically neutral in aqueous electrolyte solutions under the ambient conditions of pH ≥ 3.

Remarkably, we find that PSBMA behaves like a polyanion and moves towards the positive electrode, whereas PMPC behaves like a polycation and moves towards the negative electrode. The transport in the opposite directions, towards negative electrode for PSBMA and positive electrode for PMPC, does not occur. Even when the identity of the electrolyte solution is changed in the PSBMA experiments, the above effect is preserved, indicating a general phenomenon. Therefore, we deduce from the chemistry of these polyzwitterions that only the ionic group of the zwitterionic monomer that is further away from the chain backbone (negative charge for PSBMA and positive charge for PMPC) is visible to the external electric field and that the ionic group closer to the backbone (positive charge for PSBMA and negative charge for PMPC) is relatively invisible to the external electric field. Hence, the electrical neutrality of the ion-pair of the zwitterion is deviated resulting in the CSB.

Towards gaining more insights, we first hypothesize that this effect originates from differential counterion binding to the ionic

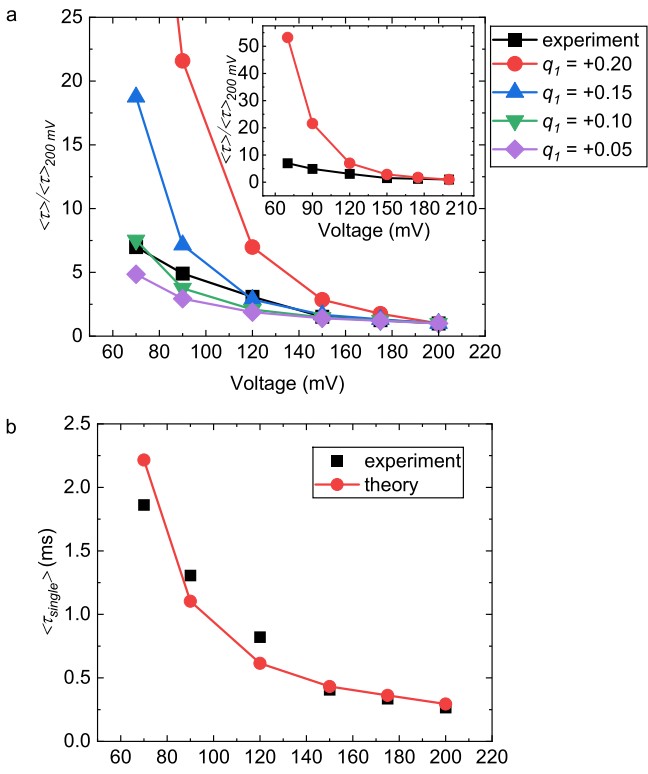

**Fig. 7 | Comparison between theory and experiment giving magnitude of CSB.** The experimental data is from Fig. 3(f). **a** Theoretical predictions of the ratio of mean translocation time at different voltages to that at the reference voltage 200 mV for different choices of $q_1$. Experimental values are black squares. Inset: expanded data sets for black squares and red circles for $q_1 = +0.20$. **b** Comparison between experimental data and theoretically predicted values for $q_1 = +0.10$.

groups of the zwitterionic monomer. This, in turn, is rationalized to arise from a positive gradient in the local dielectric constant from the chain backbone to the bulk solution. Using the Fokker-Planck formalism[66], we have calculated the average translocation time ($\langle\tau\rangle$) in terms of charge attributes of the zwitterionic groups constituting the polymer, conformational entropy, and pore-polymer interaction. Comparing theoretical predictions on the voltage dependence of $\langle\tau\rangle$ with experimental results, we have determined the charge of the inner ionic group and the magnitude of CSB. Furthermore, using counterion binding equilibrium, we have related the observed charge with the local dielectric constant, which is about 25 compared to about 70 in the bulk aqueous electrolyte solution. While this evaluation of the local dielectric constant based on our zeroth-order mean-field theory is only approximate (where the values of the parameters in the theory are not ab initio values but contain uncertainties), the observed CSB in polyzwitterions is robust and universal.

This discovery opens several new horizons. As an example, it is of interest to investigate translocation of polyzwitterions through chemically decorated pores such as $\alpha$-hemolysin and MspA protein pores towards an understanding of the interference/cooperativity from external charge decoration on CSB. Furthermore, the present results emphasize the important roles from the wavevector dependence of the dielectric constant at short length scales around polyzwitterion chains which has been overlooked in classical models for counterion binding on polyelectrolytes. It would be of great use to directly determine the dielectric function at short length scales using other experimental techniques such as fluorescence methods. Furthermore, more in-depth investigations are needed even for the present

polyzwitterionic systems that include the effects of identity of salt ion, quantification of EOF, and molecular modeling to identify polyzwitterion conformations during translocation. Furthermore, the decoupling between the dynamics of small ions and the polymer, operative at higher salt concentrations, is modified at lower salt concentrations[68–70], which might be of interest in future studies.

As another important example, it is of great significance to investigate the role of neutral and polar moieties in sequences of biological macromolecules on their transport properties in crowded aqueous environments. In particular, most of the macromolecules constituting biomolecular condensates in various membrane-less organelles are dynamic and move around in a highly heterogeneous dielectric and electrical environment under crowded conditions. The chemical sequences of these intrinsically disordered macromolecules generally consist of ionic, polar, and hydrophobic groups. So far, attempts to describe the movement of these molecules in response to local electric fields have focussed only on monopole charges on the ionic groups, and the contribution from the other nominally neutral (polar and nonpolar) groups have been presumed to be null. However, in view of the present work, the role of polar groups in the intrinsically disordered macromolecules need to be included towards a better understanding of their dynamical regulation and inter-molecular recognition.

## Methods
### Materials
PSBMA and PMPC were used as our polyzwitterions (for PSBMA, degree of polymerization (DP) = 513 and dispersity = 2.03; for PMPC, DP = 100 and dispersity = 1.40). Both PSBMA and PMPC were synthesized by RAFT polymerization in 2,2,2-trifluoroethanol (TFE) with 4,4'-azobis(4-cyanovaleric acid) as an initiator and 4-cyano-4-(phenylcarbonothioylthio)pentanoic acid as a chain transfer agent (CTA). To determine DP and dispersity of PSBMA, size-exclusion chromatography (SEC) in TFE containing 0.2 M sodium trifluoroacetate were performed at 40 °C at a flow rate of 1 mL/min using a Agilent 1200 series system, equipped with refractive index detector, PFG analytical linear M columns (8 × 300 mm, particle size 7 mm), a PFG guard column (8 × 50 mm), and an isocratic pump. DP of PMPC was determined by $^1$H NMR spectrum studies, and dispersity of PMPC was determined by using an Agilent 1260 infinite series system equipped with Ultrahydrogel linear columns (7.8 × 300 mm), an Ultrahydrogel DP matrix guard column (6 × 40 mm), and a refractive index detector, in water containing 20% acetonitrile/0.1 M sodium nitrate/0.02wt% sodium azide against poly(ethylene oxide) standards at 35 °C at a flow rate of 1 mL/min. Hydrolysis of PSBMA to methacrylic acid was evaluated by potentiometric titration (Supplementary Fig. 17a) and $^1$H NMR (Supplementary Fig. 17b), and a negligible hydrolysis was detected as lower than 0.1% (Supplementary Note 5). For PSBMA, $^1$H NMR (500 MHz, 1 M NaCl/D$_2$O, $\delta$): 4.57–4.30 (2H), 3.87–3.68 (2H), 3.65–3.49 (2H), 3.30–3.10 (6H), 3.00–2.87 (2H), 2.34–2.13 (2H), 2.13–0.74 (5H + CTA); for PMPC, $^1$H NMR (500 MHz, 0.1 M NaCl/D$_2$O, $\delta$): 7.96–7.44 (CTA), 4.30–3.95 (6H), 3.64 (2H), 3.19 (9H), 2.14–0.75 (5H + CTA) as in Supplementary Fig. 18.

### Potentiometric titration of PSBMA and PMPC solutions
Potentiometric titrations were performed on PSBMA and PMPC solutions using a METTLER TOLEDO FiveEasy pH Meter F20. For pH measurement, a small pH probe (LE422, METTLER TOLEDO) was inserted into a solution after the tube was shaken. The top of the probe was placed above the bottom of the tube, and the level of the solution was above the junction. Each pH value was measured at least 1 min after the addition of titrant and was recorded if the value stayed constant for at least 20 s. The pH of solutions were initially lowered to around 1–2 by adding 1 N HCl and then increased by adding 0.6 N or 1 N NaOH.

## Nanopore fabrication

A nanopore was made within a 10 nm-thick free-standing silicon nitride membrane on Si wafer, customized from Norcada, Canada, following the protocol described in previous works[60,61]. The protocol consists of three steps: fabrication, conditioning, and pore size measurement. In the fabrication step, both sides of the membrane were filled with 1 M KCl and 10 mM HEPES (4-(2-hydroxyethyl)-1-piperazineethanesulfonic acid, purchased from Fisher BioReagents) at pH 8. Applying linearly increasing voltage across the membrane, dielectric breakdown of the silicon nitride was detected from a sudden increase of current between chambers. In the conditioning step, the diameter of pore was adjusted by applying alternating voltages in 3.6 M lithium chloride (LiCl, purchased from Thermo Scientific), 10 mM HEPES, pH 8. In the pore size measurement step (using the same conditioning buffer), the pore diameter, $d_p$, was calculated from equation (10), where G is the pore conductance, $\sigma$ is the conductivity of the conditioning buffer (16.3 S/m), and t is the thickness of the membrane (10 nm). The final pore diameter was obtained by measuring the pore conductance (G) in the voltage range of −0.5 V to +0.5 V after 10 min of equilibration from

$$d_p = \frac{G}{2\sigma}\left(1 + \sqrt{1 + \frac{16\sigma t}{\pi G}}\right). \tag{4}$$

The pore was used only when the current *versus* voltage curve did not show rectification as an example in Supplementary Fig. 19.

## Translocation measurements

With both chambers filled with one of 1 M KCl, 10 mM HEPES at pH 7, 3.6 M LiCl, 10 mM HEPES at pH 6, or 3.6 M LiCl, 10 mM HEPES at pH 3.7, current signals were detected at constant applied voltages at room temperature (19–23 °C), using Axon patch clamp (AXOPATCH 200B) and a low-noise data acquisition system (Axon Digidata 1322A). Only if the measured current (open pore current, with only buffer solution without any polymer) was stable enough without any transient blockages for 30 s, current traces with PSBMA or PMPC in the donor chamber were then recorded. The data acquisition rate was 250 kHz, and the frequency of the low-pass Bessel filter was 100 kHz. We limited measurement time at 150, 175, 200 mV to 1–2 min to minimize pore size expansion over time under our experimental conditions, and the measurements were consistently conducted from the highest to the lowest voltage to eliminate the possibility of pore expansion affecting the increase of $R_c$ and the decrease of $\langle \tau \rangle$ with voltage. Data were analyzed only when the pore diameter remained unchanged after measurements.

## Capture rate data analysis

Analogous to the previous works[14,38,42] on single-molecule translocation, we assume that the capture of PSBMA at the pore is a Poisson process with the distribution of $\tau_a$ given as $P(\tau_a)$,

$$P(\tau_a) = R_c e^{-R_c \tau_a}. \tag{5}$$

Histograms of $\tau_a$ in the range of 0–0.6 s are drawn and then fitted with the exponential distribution (see Supplementary Figs. 20 and 21) by MATLAB (The MathWorks, Inc. Natick, MA) scripts written by Plesa and Dekker[35]. The fittings at all voltages are very good with 95% confidence interval of less than 3 Hz for $R_c$, by ignoring the first few bins.

## Translocation data analysis

$I_0$, $I_b$, and $\tau$ of each blockage event in current traces were detected by the MATLAB scripts written by Plesa and Dekker[35], where 10 kHz Gaussian filter was used, except for data with 3.6 M LiCl, 10 mM HEPES, pH 6, to reduce noise. Histograms of $I_b/I_0$ were obtained using OriginPro (OriginLab Corporation, USA), which were subsequently fitted

with multiple Gaussian distributions in the range of $I_b/I_0$ of 0–0.6 assuming blockage events with $I_b/I_0 > -0.6$ were unsuccessful translocations. The $I_b/I_0$ histograms were again fitted to another multiple Gaussian distributions in the range of $I_b/I_0$ from 0 to the highest peak position + 0.4. From the Gaussian fits, we obtained mean $I_b/I_0$ for single-file and ips translocation populations ($\langle I_b/I_0 \rangle_{single}$ and $\langle I_b/I_0 \rangle_{ips}$), respectively, and standard deviation values of $I_b/I_0$ for single-file and ips translocation populations ($STD_{single}$ and $STD_{ips}$), respectively. Taking blockages with $\langle I_b/I_0 \rangle_{single} < I_b/I_0 < \langle I_b/I_0 \rangle_{single} + 2STD_{single}$ ($\langle I_b/I_0 \rangle_{ips} - 2STD_{ips} < I_b/I_0 < \langle I_b/I_0 \rangle_{ips}$ for ips translocations), $\tau$ histograms for single-file (ips) translocations were plotted in the range of $0 < \tau < 10$ ms ($0 < \tau < 100$ ms for ips translocations).

## Statistical information

For all event scatter plots and histograms for $I_b/I_0$ and $\tau$, the value of $n$ is explicitly included in the figures. For example, $n = 2289$ in Fig. 3b. For both $I_b/I_0$ and $\tau$ histograms, $n$ is the number of events used for Gaussian fittings and log-normal fittings, respectively. In Fig. 3 for PSBMA in 1 M KCl, the data are accumulated from three independent measurements using three different nanopores to collect at least 200 blockages for one $\tau$ histogram. Each independent measurement showed the trend of $\tau$ decreasing with voltage. For PMPC translocation experiments, only electrode polarity was switched and one data set is presented in Fig. 5 to exhibit the opposite direction of translocation compared to PSBMA. We observed the same electrode polarity dependence with three different pores. In Fig. 4 for PSBMA in 3.6 M LiCl, one data set is given. Two other independent measurements were made for the ionic current traces, all of which showed the same trend for the decreasing dependence of the average translocation time with voltage. Since the diameter of the freshly fabricated nanopores in these independent measurements were slightly different from each other, but around 3.7 nm for PSBMA, error bars are not included in Figs. 3e, f, and 5d.

## Determination of charge

The experimental values of $\langle \tau \rangle / \langle \tau \rangle_{200mv}$ for different voltages (in the range of 70–200 mV) are included in Fig. 7a as black data points. For each choice of $q_1$, there is a difference between the theoretical values and the experimental value at every voltage. To find the value of $q_1$ for the best fit between experiment and theory, we define the error $\epsilon_\tau$ as $(((\langle \tau \rangle / \langle \tau \rangle_{200mv})_{theory} - (\langle \tau \rangle / \langle \tau \rangle_{200mv})_{experiment}) / ((\langle \tau \rangle / \langle \tau \rangle_{200mv})_{theory}$ and determine the mean and standard deviation (STD) of $\epsilon_\tau$ values at each $q_1$ value. For example, from the five $\epsilon_\tau$ values at each $q_1$ value with the reference voltage of 200 mV, we calculated mean $\epsilon_\tau$ and STD. We find the value of $q_1$ corresponding to the minima of the mean $\epsilon_\tau$ and STD as 0.10. By repeating this exercise for other five reference voltages (70, 90, 120, 150, and 175 mV), we find the best values of $q_1$ as 0.05, 0.10, 0.05–0.10 (0.05 from minimum mean $\epsilon_\tau$ and 0.10 from minimum STD), 0.10, and 0.10, respectively. In view of this analysis, we conclude that the best value of the charge of the quaternary ammonium group as +0.10.

## Data availability

The data presented in this study are openly available as Source Data in article. Additional data that support the findings of this study are available from the corresponding author upon request. Source data are provided with this paper.

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

## Acknowledgements

The authors are immensely grateful to Pranav Viswanathan for providing PSBMA samples, and Deborah Snyder for providing PMPC samples for our earlier experiments. It is a great pleasure to acknowledge Kuo Chen, Minglun Li, Pranav Viswanathan, and Siao-Fong Li for stimulating discussions. The authors appreciate Carla G. Steppan, Grace Leone, and Professor Todd Emrick for training Y.L. with synthesis and GPC analysis of polyzwitterions. Acknowledgment is made to the National Science Foundation (Grant No. DMR 2015935) and AFOSR (Grant No. FA9550-23-1-0584).

## Author contributions

Y.L. performed all experiments and theoretical calculations, and analyzed the data. M.M. designed the project and performed theoretical calculations. Y.L. and M.M. discussed the results and wrote the paper.

## Competing interests

The authors declare no competing interests.
