## [Peer Review file · Nature Communications]

Charge Symmetry Breaking in Neutral Polyzwitterions

Corresponding Author: Professor Murugappan Muthukumar

Version 0:

Reviewer comments:

Reviewer #1

(Remarks to the Author)

The manuscript "Electrophoretic Mobility of Neutral Polyzwitterions: Charge Symmetry Breaking" investigates the response of polar and neutral monomers in macromolecules to an electric field in their crowded aqueous solutions. This is a fundamental question of relevance in various fields such as protein behavior up to polymer separation or behavior in confined space.

The study bares the potential to be published in Nature Communications but in its current form too many open questions remain. I suggest major revisions and a subsequent review before a final decision.

More specific comments below:

1) In the introduction the authors refer to "neutral" polymers while their results are about zwitterionic polymers. This is misleading. In addition, there is literature on the non-homogeneous distribution of ions along polymer/ polyelectrolyte chains and thus on local variation of charge in such or similar charged polymers when looking at the length scale of the polymer chain length or smaller. To my opinion this should be made clear and the term "neutral" referring to the same number of positive and negative charges should be replaced by zwitterionic, as the authors investigate zwitterionic but not neutral polymers.

2) How can the authors separate a response of a solvent / counterions to the electric field from the response of the polymer? Would the conclusions change in case local charge variation would be investigated with a method not using electrochemistry?

3) In the caption of Fig. 1 the authors state: "Titration curve of PSBMA (80 mM in 1 mL of 1 M KCl) showing charge neutrality at the experimental pH range.". The "experimental pH range" is indicated at the inflection point of the curve corresponding to the pKa. This pKa is in the pH-range of 6-8.

3a) In the discussion of the Figure the authors state: "At the experimental pH 7, the polymer is neutral as evident from its pH titration curve (Figure 1b)". I do not agree to this term "neutral". I assume the authors want to imply that the number of protonated and deprotonated functional groups equals. I suggest to reflect this point and potentially correct this in the manuscript. In addition, zwitterionic polymers show complex behavior going beyond "classical" neutral polymers. That's why, to my opinion, it is a simplification specifying a zwitterionic polymer as "neutral". For example it has been reported that zwitterionic polymers included into a nanopore can result into so-called "bipolar" pores, excluding positive and negatively charged ions. See e.g. https://pubs.acs.org/doi/epdf/10.1021/ac00065a007?ref=article_openPDF
https://pubs.acs.org/doi/epdf/10.1021/ja8086104?ref=article_openPDF

So maybe the term "neutral" polymer should be modified.

3b) In addition, to my understanding of the molecular structure and behavior of NR₄ and R-SO₃⁻ groups, I would expect that only the R-SO₃⁻ group is protonated or deprotonated at this pH-range. I would expect the NR₄ group to be permanently charged within the applied pH-range. If I do not miss any information the statement "charge neutrality" is misleading? This would then rather imply more positive than negative charges. When further following the authors explanation referring to stronger / weaker ion binding, this would then emphasize the ion effect even more? I can understand the argument of stronger ion binding to permanent positive charges, screening charges on the cost of the entropy of counterions and finding an equilibrium with polymer chain stretching on the cost of conformational entropy. This phenomenon is e.g. known from polyelectrolytes and can be investigated for example by measuring the ion activity through osmotic pressure. This statement should be reflected and specified or corrected. In addition, could the authors add the changes in molecular structure with going from acidic to basic pH into Figure 4a and Figure 1a please? Maybe a zetapotential measurement would be helpful to clarify this point, too.

4) Figure 1 g and h shows the molecular structure and the dielectric constant along the molecule length (r in nm). It would be

helpful if the values of r (x-axis Figure 1h) are indicated in the scheme of Fig. 1g.

5) Further the authors state at page 4: "Thus, PSBMA behaves like a polyanion even though it is neutral at the experimental pH (Figure 1b). This means that the charge symmetry between positive and negative charges constituting the zwitterionic group is broken. We show below that this unexpected phenomenon of charge symmetry breaking (CSB) is universal for all polyzwitterions independent of the zwitterion's dipole orientation to the". This conclusion may be reconsidered based on the above mentioned charge situation and ion effect. I'm not convinced that the interpretation that the number of positive charges equals the number of negative charges at the pKa value is globally correct for the investigated zwitterionic molecule.

6) For PMPC in Figure 4 it seem not surprising to me that this behaves like a cation assuming that at pKa 50% of the phosphate is protonated and 50% deprotonated and NR4 is permanently positively charged. Please reflect on the protonation-deprotonation equilibrium of both of these groups with pH and if needed adapt the discussion. Please add the molecular structure for acidic and basic pH of Figure 4a.

7) To further support the authors mechanistic explanation, I suggest to repeat the experiment not at the pH equal to the pka but at a pH at which one group should be fully protonated or fully deprotonated.

8) Regarding the translocation experiments: Is the silicon nitride nanopore charged and if yes how does the nanopore charge affect the result? What is the length and shape of that nanopore?

9) At page 5 the authors state: "Therefore, in view of equation (1), the ionic group of the zwitterion closer to the backbone is subjected to more counterion binding compared to the ionic group further away from the chain backbone.". The authors may comment on the influence of ion type e.g. referring to contact ion pairs or water mediated ion-pairs for example.

10) Do the authors have any information on the ion-charged group interaction on the molecular level e.g. by MD simulation?

Reviewer #2

(Remarks to the Author)

This work investigates translocation of polyzwitterions through solid state nanopores. There are some interesting observations, such as the reversal of direction when the chemical structure is reversed. However, the qualitative explanation for the observed phenomenon lacks independent validation and the quantitative predictions have too many uncertainties. Additionally, even if revisions are made I think this work could be more suitable for a journal dedicated to macromolecules, such as ACS Macromolecules.

The main claim of this work is that polyzwitterions exhibit electrophoretic mobility. However, only resistive pulse sensing with nanopores is used for investigation. It is well known that the translocation through nanopores is not governed only by electrophoresis, also electroosmosis and dielectrophoresis come into play:

<https://dx.doi.org/10.1021/acsnano.0c06981>

How can the authors exclude these other effects? There is a lack of complementary techniques to verify the electrophoretic mobility although it is indicated by the linear relation of voltage and event rate. It would be very simple to test mobility under a field in bulk solutions. The molecules can be labelled if need be. In fact, it is hard to understand why the nanopore system was chosen to begin with if the purpose is to investigate electrophoretic mobility.

I also think that even for the nanopore measurements alone, the study could be extended to see what happens at lower ionic strength or different pH, and by using zwitterions with weak acid or base groups. These are quite obviously important parameters to vary and would shed light on the mechanisms. Complementary experiments on polyelectrolytes would also be very useful.

As far as I understand the main physical explanation for the findings is related to a varying degree of counterion condensation on the outer vs inner groups. This is reasonably well described in the "Charge regulation by counterion binding" section. However, some wording is still misleading. Why is "degree of ionization" discussed? Both the negative and positive charges are strong and always ionized. I also think the naming "charge symmetry breaking" is confusing. The distribution is still cylindrically symmetric?

The theoretical model of translocation time is quite detailed but a bit cumbersome to follow for a non-specialist. I believe it could be shortened in the main text and extended in a supporting information section. More importantly, there are many unknown parameters in this model. It appears that the purpose is to get an independent estimate of translocation time to be compared with experiments so that the charge of the translocating species can be obtained. However, my impression is that there are too many uncertainties or vaguely motivated values (e.g. interaction energy with pore wall) to trust the outcomes quantitatively. In particular, a specific value of the degree of counterion condensation on the outer groups is assumed (0.25) and this seems rather arbitrarily, yet it is critical for obtaining the degree of counterion condensation on the inner groups. Shape asymmetry may also influence translocation directionality. Do the pores exhibit rectification? The study should include characteristic IV curves and as a control experiment, it should be verified that translocations occur (or do not occur) when the voltage is reversed over the same pore and molecules introduced on the other side.

I cannot see how the folded events are separated from the unfolded ones. Their distributions are clearly overlapping (Fig. 3b). Also it seems that unfolded events may be missed based on the distribution of dwell times (Fig. 3d). What is meant by "most of those events show two different blockage levels" in the quantitative sense?

The threshold behavior in voltage is speculative. Why not perform measurements at lower voltages to properly investigate? If N and M denote number of monomers, not length, dividing the voltage with these numbers does not give a field.

I believe the titration curves are not very interesting and can be moved to supplementary information.

Fig. 3f should not be needed as it repeats information.

The structure of the paper is strange because central results are presented already before the results section.

Both the introduction and discussion sections contain strange comments on the connection to biological system. I do not see

the relevance since transport in cells does not occur by electric fields.

Page 19: "the local translocation speed of the monomers of the polyzwitterion is essentially uniform at all voltages in the present study" Please clarify, if the dwell times vary the speed of the monomers must also vary?

Page 20: "wavevector dependence of the dielectric constant" What waves?

Reviewer #3

(Remarks to the Author)

The manuscript by Lee and Muthukumar studies the important topic of the translocation characteristics of polar and neutral polymers through nanoscale pores. There has been a lot of work on understanding the passage of uniformly charged polymers through nanopores for DNA sequencing applications, but much less work on so called zwitterionic polymers. This is of growing interest with the emergence of new applications nanopores for proteomics and glycomics. The authors proceed with studying two simple polymers [poly(sulfobetaine methacrylate) (PSBMA) and poly(2-methacryloyloxyethyl phosphorylcholine) (PMPC)] as model systems for the behaviour of zwitterionic polymers. This type of work is important in order to develop a better understanding of their passage dynamics and thus detection strategies for improved sensing performance. The authors find that the charge of the zwitterionic group farthest from the backbone dominates the electrophoretic response of the polymer. This is attributed to less shielding of this ionic group of the zwitterionic monomer. The authors refer to this differential counterion binding phenomenon as a charge symmetry breaking (CSB).

Although the topic and ideas introduced in the manuscript are of high interest, the results presented in the manuscript in its current form do not fully support the conclusion. More experimental work and/or analysis is needed before it can be considered for publication. At the moment, the interpretation of the observed nanopore signals, and the experiments shown do not sufficiently support charge symmetry breaking as being the phenomenon responsible for the passage characteristics.

Below is a list of major points that need to be addressed:

- The manuscript would benefit greatly from a clearer physicochemical description of the polymers used in its study, i.e. polymer length, cross-sectional diameter or persistence length. These parameters are crucial for helping readers interpret nanopore translocations signals and relating them to the conformations held during passage. Namely, Figures 3b and 3c show single-file blockages centered around $l_b/l_o \approx 0.6$ which, to first order, suggests a polymer cross-section diameter of $d \approx d_{\text{pore}} \sqrt{(\Delta/l)} \approx 2.5\text{nm}$. Is this the value expected for an elongated polymer, or does this indicate that polymers are more blob-like inside pores? Given that polymers being stretched inside pores is a central assumption of the theoretical model, experimental data should be used to better interpret polymer conformations during translocation. For example, ssDNA does not produce quantized blockages unlike dsDNA because of their very different persistence length. Why are these polymers expected to translocate as single-files with some folds and not as blobs?
- The interpretation of single-file and folded translocations should be elaborated more or simply revisited. Namely, are events interpreted as folded actually folded or correspond instead to two polymers inside the pore at once? The latter seems plausible given the similar timescale of dwell times and inter-event times: Figure 3 shows that at 150mV $\langle \tau \rangle \approx 7.5$ ms and $\tau_a = 1/60\text{Hz} = 17$ ms. Do folded translocations always begin with a deeper blockade, as expected from polymers bending inside the pore? This should also be clarified.
- It should be further noted that an exponential dependence of translocation time on voltage is not common for electrophoretically-driven translocations which instead commonly result in inverse voltage relationships: $\tau \sim \Delta V^{-1}$. Exponential relationships have instead been observed when a molecule needs to overcome a strong energy barrier to enter pores, such as for DNA hairpins through biopores or translocations of long self-hybridized ssDNA through solid-state nanopores.
- Strikingly, the argument that PMPC translocates towards the negative electrode as a polyanion is not a convincing proof of Charge Symmetry Breaking, given that electroosmotic flow (EOF) arising from negatively charged SiN also flows towards the negative electrode. Additionally, the section addressing PMPC translocations would highly benefit from a more in-depth analysis like the ones shown for PSBMA, i.e. capture rate and translocation time vs voltage. In addition to EOF surprisingly never being discussed as a source of variation for the CSB phenomena, other important physical phenomena were dismissed. For instance, the electric field inside a nanopore is non-uniform, and is strongest towards the pore walls, i.e. towards the ionic groups further away from a stretched-out chain backbone.
- The content of this research is very nanopore-centric which, as discussed, can introduce forces and phenomena specific to nanopores, as opposed to the more general electrophoretic mobility of neutral polyzwitterions, as suggested by the title. This research and its conclusions could also benefit from running gel electrophoresis experiments to support its conclusions and interpretations.
- The authors write on page 2: "even the simplest limit of homopolymers made of only neutral groups (so that complications from sequence effects can be deliberately suppressed) is yet to be investigated." But a lot of work has been done studying the passage of PEG of different molecule weight through nanopores that are neutral or very weakly charged as well as proteins and peptides so that statement should be revised.
- Performing more experiments while changing the pH of the solution would really provide more results to better understanding the regime of passage (CSB dominated or not), by changing the protonation state of the polymer and also of

the pore wall (affecting EOF).

Other minor comments

- Each captions should include the experimental parameters used to acquire the results, pore size, voltage, concentration of analyte, of salt, sampling rate, low-pass filter value, etc...
- Capture rate data should be reported in Hz/nM.
- With a >40Hz capture rate, why show plots with only <4000 events, this represents only 100 seconds of recording. It is statistically enough, but nanopore experiments are often carried out for longer.
- Why is the number of events changing in the various panels of figure 3?
- Which experimental data is the model compared to in Figure 7? The authors should also compare with the LiCl data, which is known to shield more.

Version 1:

Reviewer comments:

Reviewer #1

(Remarks to the Author)

All comments have been fully addressed, to my opinion. I recommend to publish the manuscript in its revised present form.

Reviewer #2

(Remarks to the Author)

While the revised version addresses some of my comments in a satisfactory manner, others remain unresolved in my opinion and the authors have chosen not to perform many of the suggested experiments and corrections. As I mentioned, there are certainly interesting aspects of this work but I still cannot recommend that it is published (at least not in Nature Communications) unless additional work is performed to properly understand the proposed mechanism.

Starting with the good news, the authors have addressed my (and other reviewers) comments on the role of electroosmosis and dielectrophoresis good enough. I agree that the electroosmotic force will be opposing and that dielectrophoretic effects should at least not exhibit directionality. The complementary IV curve is also appreciated and several clarifications have been done.

My main remaining concern is that the electrophoretic mobility of the zwitterions, which must be regarded as the most central result in this work, has still not been investigated by any complementary method. The authors claim that this would not be relevant since the polymer would not assume a linear conformation in bulk solution. I fully agree, but neither do they in the reservoirs outside the nanopores, and yet they are attracted to the pore, apparently by electrophoresis. As I'm sure the authors are aware, the behavior they observe (frequency is linear with voltage) is established to be due because of drift motion (dominating over diffusion) within a certain distance (several microns I believe) away from the pore (Wanunu et al. Nat. Nanotech. 2010 and others). This means that the polymers seem to be moving in the field outside the pore, where they clearly must be random coils (as drawn by the authors). Therefore it should be possible to observe such motion also with conventional electrophoresis methods, capillary or gel, contrary to what the authors claim. Nanopore sensors are in the end very complex systems that are not designed to investigate electrophoretic mobility. We have other established tools for this. I emphasize that such measurements are important because either way they will show that the current interpretation is not correct, or at least incomplete: If mobility is observed, it means that the charge symmetry breaking is not sufficient to explain electrophoretic mobility (as it is based on a linear conformation). If mobility is not observed, there must be another explanation why the polymers are driven towards the pore.

Related to the above, it is well-known from many studies using zeta potential measurements that the effective charge of macromolecules may be counterintuitive. For surface tethered polymers, negative charges have been observed on "neutral" polymers (example Zimmerman et al. Langmuir 2005, 21, 5108-5114) by streaming potential. These effects are likely related to chemical affinities between certain ionic species and organic groups. The authors have performed no such characterization. Measurements can of course also be done in solution phase by phase analysis light scattering. This also illustrates why measurements at different pH may be very informative as even non-ionizable polymers may alter their electrokinetic properties, but the authors have (still) not performed any such tests.

Regarding the model and values of different parameters, I still see several issues. The authors write that Supplementary Note 3 should explain why their choice of values is reasonable, but I find no such motivation in that section. The value of 0.25 for the effective charge of the outer groups of the zwitterion is apparently taken from a study on DNA, i.e. quite a different molecule. The interaction energy between monomer and wall is estimated under the assumption that only electrostatics contribute. I doubt this is the case and (again) it would be possible to perform experiments with other techniques to obtain more information on interactions between the zwitterions and silicon nitride surfaces. These points raise the question whether the agreement between theory and experiment in Fig. 7b is simply a result of too much freedom in multiparameter tuning. To be fair, this may not be a major flaw, but I think the authors need to be more open with the fact that the uncertainty is high with respect to the values they pick.

For the rest, I have no more comments that require addressing by absolute necessity. I just leave some more thoughts here for the authors that they can consider (or ignore if they wish).

- I find it unlikely that most readers will prefer to see results and discussion in the introduction part of a paper (before the Results section).
- I still do not see any clear relevance to biological systems. Like the authors say, in a cell there are plenty of charged molecules in a "soup" - where would there be directional mobility in a field? (An exception would be the potential across certain membranes but I still do not see any relevance to zwitterions in such scenarios.)
- I'm sorry for "stunning" the authors by asking what wavevector they were referring to. I do know that the dielectric constant has a frequency dependence, but I still do not understand what the authors are trying to say. Are there electromagnetic waves coming into play in this work?

Reviewer #3

(Remarks to the Author)

We want to thank the authors for their detailed and transparent responses to the comments from the different reviewers. They have performed new experiments and added new valuable discussions, which have resulted in a strengthened version of their manuscript. Of particular note, the new experiment recording PMPC in pH 3.7 solution now removes any doubt of EOF being the main mechanism responsible for driving capture and translocation of the polyzwitterion through the pore and thus strengthens the case for the charge symmetry breaking phenomena introduced.

Nonetheless, we feel that two points should still be better addressed. The CSB mechanism is sound, and well described, but the manuscript would significantly benefit from having clearer presentation and interpretation of some aspects of the data:

- Nature of the signals. As of now, it is not clear what the translocation signals correspond to, i.e. what conformations are held by the polymers as they pass through the pore in the single-file or folded states. This is an important point, especially because of the theoretical discussion and derivations at the end of the manuscript, and should be presented more clearly:
- On page 16: "The cross-sectional diameter that can be qualitatively inferred from the ratio of blocked current in single-file blockages to the open pore current is ~2.9 nm, consistent with the value obtained from molecular dynamics." How was the 2.9 nm value obtained? Although 2.9 nm is only 30% off from the expected 2.2 nm diameter, i.e. $(2.9-2.2)/2.2$, the current blockages which roughly scale with the squared diameter would be off by 74%: $(2.9^2-2.2^2)/2.2^2$. This significant deviation should be addressed.
- The translocation times are quite long for such small molecules. Are the molecules expected to be in a stretched conformation, i.e. under tension, when inside the pore? As a rough reference, how do translocation times compare to the relaxation times?
- Given the above points and the low persistence length of the polymer, it is not clear that the polymer conformation is completely stretched out inside the pore. Even for the moderate "single-file" population, without being a blob the polymer conformation could easily be deflecting off the pore walls, i.e. not perfectly aligned with the pore axis. This locally angled conformation would result in deeper blockages, which might explain the 2.9 nm vs 2.2 nm difference. This would technically change the theoretical parameters of the models, but to first order wouldn't change conclusions.
- The discussion regarding the folding interpretation is frankly a little vague and not visually supported by the data in the Figures. Where do the folded states occur and for how long? A complete understanding or quantitative description is not necessary, but a simple suggestion would be to add a figure of many (>20) current traces of translocation events to show the reader what events look like and show where and how long folded states are. These polymer translocations are not well known by the community, and a visual demonstration of signals would be very helpful in better understanding the data, and accepting the given interpretations.
- Unlike folded dsDNA blockades, blockage levels are far from being quantized ($\Delta I_1=0.6, \Delta I_2=0.8$). As discussed, this instead suggests local kinks in the polymer conformations, and not necessarily folds, which might not be too surprising given how flexible the polymer is with its 2 nm persistence length. Maybe the authors need to define folding (we assume it is a complete fold of the polymer chain on itself, thus producing quantized levels) and if they assume such kinks are folds (which we would instead define as in a slight blob state, but not fully stretched).

2. Lack of PMPC results.

- Using PMPC as a proof of the universality of CSB is great, however the analysis of PMPC results is too minimal and binary.
- As done with the PMBSA, a simple comment on the voltage-dependence of translocation time would demonstrate that polymers are actually translocating through the pore, as opposed to simply colliding with it. Such a comment was already made in a response to one of our comments but omitted from their revised manuscript.

Minor points:

- 1. On page 5, the scale for r in [nm] in Figure 1g is confusing. Are the dielectric constant values reported for different distances? In that case it should be in [Angstroms]. h is not introduced in the caption

- On page 7, PMPC is mentioned but has not been introduced or defined yet.
- For overall readability: Subjectively, the introduction of the article “summarizes” the results of PMBSA in too much details, which makes for a repetitive read when going through the first pages of the results section (page 7).
- The population separation for moderate and deep populations is much better visualized with LiCl data (Figure 8 in SI) or 70 mV and 90 mV 1M KCl data sets (Figure 4 in SI). Why not use these in Figure 3c instead of the barely differentiated populations of 150 mV? It would help the reader reach the same conclusions as the authors without making them dig and scroll through the SI.

Martin Charron & Vincent Tabard-Cossa

Reviewer #4

(Remarks to the Author)

Version 2:

Reviewer comments:

Reviewer #2

(Remarks to the Author)

My comments after the second revision round remain largely unchanged because the authors have (once more) chosen to not perform the additional experiments I requested. Again, I do think this work has interesting aspects to it, but I also believe the results and conclusions are not solid and consistent enough to warrant publication in Nature Communications. I think we can respectfully agree to disagree that the mechanism is “clearly validated without any doubt”. I will give a few additional comments, but overall I feel like I’m repeating myself and I guess it will not change anything. I suppose the decision whether to publish the paper anyway is up to the editor.

In their previous reply, the authors argued that the CSB (and the electrophoretic mobility) is associated with the linear configuration of the chain enforced by the pore:

“In bulk solutions, a dispersed flexible polymer such as PSBMA or PMPC is not a rigid sphere, a rigid cylinder, or a rigid ellipsoid ... The net outcome is that these orientations and chain conformations are averaged out ... Such a conformational status of the molecule prevents the distinction between the ionic groups on the repeat unit in terms of their normal distance from the chain backbone ... Therefore, we designed our experiment by trying to transport the molecule through a nanopore as an unfolded single file.”

After this I pointed out that the data suggests that the chains exhibit electrophoretic mobility also in the bulk solution due to the linear capture rate with voltage. Now it seems that the authors have changed their mind:

“Of course, CSB is present even in bulk solution. Otherwise the neutral dipolar polyelectrolyte would not move under an electric field to be captured by the nanopore ...”

It’s perfectly fine to change opinion, but (as I mentioned) this means that the whole interpretation of the results and the main point of the paper makes no sense. Based on the authors’ own description of the phenomenon, how can CSB occur for the randomly oriented and fluctuating chain? Why would it be captured by a field when in the coil state where all orientations are “averaged out”? The first step to be able to answer this critical question is to investigate by a simple independent method if there really is electrophoretic mobility in bulk or not.

Regarding my request to test zeta potentials, I agree that extraction of an accurate zeta potential value is not simple for a hydrated chain, but that is not the point. The measurements (PALS) can still be used to investigate field-induced mobility. The measurements are extremely simple to do by PALS and it should be known how the instruments calculates a zeta potential based on the measured mobility.

I was also referring to streaming potential measurements on surface films. Again, these have revealed that organic materials believed to be neutral actually exhibit surface charges due to chemical interactions with ions. Another example:

<https://doi.org/10.1021/jp004051u> There are also numerous examples of polyelectrolytes which exhibit unexpected charge behaviour depending on the electrolyte. (I’m not sure why the authors discuss issues with single molecule measurements here, there would be no need to perform them on the single molecule level.)

Finally, I point out once more that experiments on surface interactions between the polymers and silicon nitride would provide further important information. Again, these would be extremely simple to perform with established methods.

Reviewer #3

(Remarks to the Author)

We thank the authors for addressing our latest sets of comments. We now support publication in Nature Communications. The added interpretation (text and schematics) of their polymer conformations will help the readers understand better the

nature of the signals observed. We look forward to seeing more results on PMPC molecules and other polyzwitterion polymers in future studies.

Reviewer #4

(Remarks to the Author)

Response to Reviewer #1

We thank the reviewer for appreciating the importance of the reported results and for excellent suggestions to enhance the manuscript. We have addressed all suggestions by the reviewer as detailed below and incorporated in the revised version and SI. The comments of the reviewer are in black and our response is in blue.

The manuscript “Electrophoretic Mobility of Neutral Polyzwitterions: Charge Symmetry Breaking” investigates the response of polar and neutral monomers in macromolecules to an electric field in their crowded aqueous solutions. This is a fundamental question of relevance in various fields such as protein behavior up to polymer separation or behavior in confined space. The study bares the potential to be published in Nature Communications but in its current form too many open questions remain. I suggest major revisions and a subsequent review before a final decision.

Response: We thank the reviewer for appreciating the fundamental significance of our results and their relevance to diverse fields. We thank the reviewer for raising several excellent questions and we have addressed all of these as detailed below.

More specific comments below:

1) In the introduction the authors refer to “neutral” polymers while their results are about zwitterionic polymers. This is misleading. In addition, there is literature on the non-homogeneous distribution of ions along polymer/ polyelectrolyte chains and thus on local variation of charge in such or similar charged polymers when looking at the length scale of the polymer chain length or smaller. To my opinion this should be made clear and the term “neutral” referring to the same number of positive and negative charges should be replaced by zwitterionic, as the authors investigate zwitterionic but not neutral polymers.

Response 1.1: We thank the reviewer for these two important comments.

Regarding the first comment, we agree that we must distinguish between our zwitterionic monomers in their electrically neutral state from the genuine neutral monomers which do not possess any charged functional groups inside them. As noted by the reviewer, we deal with the former in the present study. Accordingly,

we have explicitly spelled this out in the title, on page 2, and throughout the paper, as “neutral polyzwitterions”.

Regarding the second comment on the non-homogeneous distribution of ions along polymer length, we thank the reviewer for the opportunity to add more detailed description of the phenomenon of counterion binding. The classical model for this phenomenon in polyelectrolytes is the Manning condensation model. In this model, a certain number of ionized groups along the chain contour are neutralized (equivalently subjected to condensation) by the counterions of those ionized groups. This results in the non-homogeneous distribution of condensed counterions, or equivalently, non-homogeneous charge distribution along the chain contour. As the reviewer noted, there is considerable literature on this phenomenon. In all treatments of this phenomenon, the dielectric constant of the medium, where counterion binding occurs, is assumed to be uniform and single-valued throughout the medium. In contrast with this traditional paradigm, our experimental findings suggest that the dielectric constant in our system is not a single-valued uniform value but that there exists a positive gradient in the local dielectric constant away from the chain backbone. This in turn results in the observed charge symmetry breaking in the neutral zwitterionic repeat units of the polymer. The above arguments are now clearly explained in the revised version on pages 2-3.

2) How can the authors separate a response of a solvent / counterions to the electric field from the response of the polymer? Would the conclusions change in case local charge variation would be investigated with a method not using electrochemistry?

Response 1.2: In our view, this is an excellent remark. In fact, any charged polymer in a solution is not alone, but always surrounded by its counterions. It is well known that the dynamics of the counterions (and solvent) is strongly coupled to the dynamics of the polymer in the absence of any additional salt in the solution. This coupling is manifest in the phenomenon of ‘ordinary-extraordinary transition’ as observed in dynamic light scattering (DLS) experiments [Förster and Schmidt, 1992 (Ref.67); Muthukumar, PNAS, 2016, Ref.68]. Theory and DLS experiments (which do not involve electrochemistry) addressing this phenomenon have unambiguously established that the coupling between the counterions and the polymer is completely

broken at higher concentrations of added salt (such as 1M KCl used in the present study) [Muthukumar, 2023, Ref.7]. This decoupling is also valid even in the presence of electric force in the linear response regime, as shown in our previous work [Singh, S. P., & Muthukumar, M., 2014, Ref.69]. In view of this, we are certain that the responses from the counterions and the polymer are decoupled. Thanks to the reviewer's remark, we have briefly mentioned these remarks on page 22.

3) In the caption of Fig. 1 the authors state: "Titration curve of PSBMA (80 mM in 1 mL of 1 M KCl) showing charge neutrality at the experimental pH range.". The "experimental pH range" is indicated at the inflection point of the curve corresponding to the pKa. This pKa is in the pH-range of 6-8.

Response 1.3: We respectfully disagree and we hope that we did not mislead the reviewer. The inflection point referred to by the reviewer is not pK_a , but the isoelectric point pI . An explanation of the correct interpretation of this inflection point may be found in standard textbooks. An illustrative example is given in Figure 3-10 on page 83 of the book "Lehninger, Principles of Biochemistry, Fourth Edition" by D.L. Nelson and M. M. Cox. For easy reference, this figure is reproduced here as FIG. 1.

Analogously, in our polyelectrolyte systems of PSBMA and PMPC, with their titration curves given in Figs. 1b and 5b in the main manuscript, the vertical lines denote the isoelectric points. Due to the strong acidity of sulfonic acid group ($pK_a \sim -7$) and phosphoric acid group ($pK_a \sim 1$), all of these groups exist as deprotonated at our experimental pH (which is higher than the corresponding pK_a values by at least two), and the quaternary ammonium group (NR_4) carries a permanent positive charge. Thus, the overall net charge of the zwitterion moiety is zero at $pH \geq 3$. We apologize for not having adequately explained these details in the original version and we thank the reviewer for the opportunity to be more clear and to use the correct meaning of "neutral" zwitterionic repeat unit of the molecule. This issue has been made clear now on pages 2-3, 4, 6, and 15 in the revised manuscript.

3a) In the discussion of the Figure the authors state: "At the experimental pH 7, the polymer is neutral as evident from its pH titration curve (Figure 1b)". I do not agree to this term "neutral". I assume the authors want to imply

FIG. 1: pH titration curve of 0.1 M glycine. (copied from Lehninger, Principles of Biochemistry, Fourth Edition” by D.L. Nelson and M. M. Cox, page 83.) At the isoelectric point (pI=5.97), the chemical structure of glycine is a neutral zwitterion as shown at the top of the figure.

that the number of protonated and deprotonated functional groups equals. I suggest to reflect this point and potentially correct this in the manuscript. In addition, zwitterionic polymers show complex behavior going beyond “classical” neutral polymers. That’s why, to my opinion, it is a simplification specifying a zwitterionic polymer as “neutral”. For example it has been reported that zwitterionic polymers included into a nanopore can result into so-called “bipolar” pores, excluding positive and negatively charged ions. See e.g. https://pubs.acs.org/doi/epdf/10.1021/ac00065a007?ref=article_openPDF https://pubs.acs.org/doi/epdf/10.1021/ja8086104?ref=article_openPDF So maybe the term “neutral” polymer should be modified.

Response 1.3a: Please see Response 1.3. We fully agree with the reviewer that zwitterionic polymers are not the “classical” neutral polymers and we have now clarified this point on page 2 in the revised manuscript.

3b) In addition, to my understanding of the molecular structure and behavior of

NR₄ and R-SO₃⁻ groups, I would expect that only the R-SO₃⁻ group is protonated or deprotonated at this pH-range. I would expect the NR₄ group to be permanently charged within the applied pH-range. If I do not miss any information the statement “charge neutrality” is misleading? This would then rather imply more positive than negative charges. When further following the authors explanation referring to stronger / weaker ion binding, this would then emphasize the ion effect even more? I can understand the argument of stronger ion binding to permanent positive charges, screening charges on the cost of the entropy of counterions and finding an equilibrium with polymer chain stretching on the cost of conformational entropy. This phenomenon is e.g. known from polyelectrolytes and can be investigated for example by measuring the ion activity through osmotic pressure. This statement should be reflected and specified or corrected. In addition, could the authors add the changes in molecular structure with going from acidic to basic pH into Figure 4a and Figure 1a please? Maybe a zetapotential measurement would be helpful to clarify this point, too.

Response 1.3b: Please see Response 1.3.

At the risk of repetition of the explanation given in Response 1.3, for PSBMA and PMPC, since each of their repeat units has one permanent positive charge and one strong acidic group, we assume that all chemical structures of their repeat units are electrically neutral within the intermediate pH range around pI. This is consistent with the two references kindly quoted by the reviewer: pH dependence of sulfobetaine (strong acid and RN₄⁺)-coated column with inorganic solute was not considered in the reference (Anal. Chem. 1993, 65, 2204-2208), while pH dependence was considered for zwitterions with weak acid and weak base as in the reference (J. Am. Chem. Soc. 2009, 131, 2070-2071). By stronger counterion binding to NR₄ group of PSBMA (compared to that to R-SO₃⁻ group of PSBMA), we meant stronger ion-ion interaction in the local environment around the NR₄ group due to the proximity of NR₄ group to the oil-like (low dielectric constant) chain backbone in the chemical structure of PSBMA. The local low dielectric constant around NR₄ group of PSBMA leads to stronger NR₄ -counterion interaction, inducing higher probability of NR₄ group being neutralized.

As the reviewer has pointed out, the counterion binding is well known for polyelectrolytes leading to lower effective charge of a polyelectrolyte molecule than its intrinsic charge from its chemical structure. Thus, there must be counterion binding to R-SO₃ groups, as well. However, due to different values of local dielectric constant in the vicinity of the ions, the number of NR₄ groups neutralized is larger than that of R-SO₃ groups in PSBMA, thus leading to negative effective charge of PSBMA. Similarly, for PMPC, since the phosphoryl group is closer to the chain backbone compared to the NR₄ group, the counterion binding to the phosphoryl group is stronger than that to the NR₄ group, which makes PMPC to behave like a polycation. Such intricate features of counterion bindings near the molecule's backbone cannot be gleaned from techniques such as zeta potential measurements (where the whole molecule is averaged into a colloid-like spherical object for data interpretation). In fact, one of the main aims of the present work is to interrogate the local details of counterion binding around the chain backbone. We have now clearly elaborated these details in the revised version on pages 2-4. As suggested by the reviewer, we have included chemical structures into Figures 5a and 1b.

4) Figure 1 g and h shows the molecular structure and the dielectric constant along the molecule length (r in nm). It would be helpful if the values of r (x-axis Figure 1h) are indicated in the scheme of Fig. 1g.

Response 1.4: We thank the reviewer for this great suggestion. We have now revised Figure 1g following the suggestion. However, we note that the profile of dielectric constant along r is just a sketch, as described in our manuscript.

5) Further the authors state at page 4: "Thus, PSBMA behaves like a polyanion even though it is neutral at the experimental pH (Figure 1b). This means that the charge symmetry between positive and negative charges constituting the zwitterionic group is broken. We show below that this unexpected phenomenon of charge symmetry breaking (CSB) is universal for all polyzwitterions independent of the zwitterion's dipole orientation to the". This conclusion may be reconsidered based on the above mentioned charge situation and ion effect. I'm not convinced that the interpretation that the number of positive charges equals the number of negative charges at the pK_a value is globally correct for the investigated zwitterionic molecule.

Response 1.5: This is already addressed in the above responses. In particular, please see Response 1.3. To avoid any potential misunderstanding by the readers, we have now clearly described this issue throughout the revised manuscript. In particular, we have explicitly stated by adding the sentence “Here, we consider strong polyelectrolytes as neutral based only on their chemical structure’s electrical charges determined by protonation/deprotonation.” on page 2 of the revised manuscript.

6) For PMPC in Figure 4 it seem not surprising to me that this behaves like a cation assuming that at pK_a 50% of the phosphate is protonated and 50% deprotonated and NR4 is permanently positively charged. Please reflect on the protonation-deprotonation equilibrium of both of these groups with pH and if needed adapt the discussion. Please add the molecular structure for acidic and basic pH of Figure 4a.

Response 1.6: We have already addressed this issue in Responses 1.1, 1.3a, and 1.3b. Nevertheless, we would like to acknowledge the care and scholarship of the reviewer.

7) To further support the authors mechanistic explanation, I suggest to repeat the experiment not at the pH equal to the pK_a but at a pH at which one group should be fully protonated or fully deprotonated.

Response 1.7: Please see our responses, Response 1.3a and Response 1.3b.

8) Regarding the translocation experiments: Is the silicon nitride nanopore charged and if yes how does the nanopore charge affect the result? What is the length and shape of that nanopore?

Response 1.8: Yes, the silicon nitride nanopore is widely known to be weakly charged. There are two kinds of consequences of the surface charge on translocation kinetics of polyelectrolytes.

First is the electrostatic interaction between pore surface charges and charged groups of polyelectrolytes, which we have already described as F_{pore} in the original manuscript. The exact value of the surface charge density for our experimental conditions (pore diameter, salt concentration) is not known. Even though several different values for surface charge density are reported in the literature, the values are so small that the calculated values of F_{pore} are negligible compared to the other contributions. As a specific example, by taking even a high value of -0.12 C/m^2 at pH 7 (as reported in Ref. 64), the calculated value of F_{pore} is negligible, as already

detailed in our original Supplementary Note 4 (7 in the revised manuscript).

The other consequence of surface charges is the electroosmotic flow (EOF), Silicon nitride nanopores carry a weak negative charge density in our experiments at pH 7. This generates EOF of the fluid inside the nanopore towards the negative electrode. In the presence of this EOF, the electrophoretic translocations of PS-BMA (towards positive electrode) and PMPC (towards negative electrode) occur. Stimulated by a remark from the third reviewer and additional experiments, we find that EOF is weaker than the electrophoretic force in determining the translocation direction. We have now provided a detailed discussion of EOF on pages 4, 7-8, 13, 16 in the revised version as well as in the detailed response to the third Reviewer.

As already mentioned in the original version, the length of the pore is 10 nm as provided by the vendor, and the pore shape is roughly cylindrical as expected in typical pore formation using the controlled dielectric breakdown protocol (Ref.60). This point is now mentioned in Supplementary Note 3.

9) At page 5 the authors state: “Therefore, in view of equation (1), the ionic group of the zwitterion closer to the backbone is subjected to more counterion binding compared to the ionic group further away from the chain backbone.”. The authors may comment on the influence of ion type e.g. referring to contact ion pairs or water mediated ion-pairs for example.

Response 1.9: This is a stimulating remark and the whole community is far behind in understanding the role of the ion type. Even restricting to monovalent counterions, the binding between ions is controlled by many effects such as entropy associated with water reorganization when a pair of ions approach each other, extent of loss of translational entropy of counterions when they bind, self-consistent conformational changes of the polymer molecule due to binding, polarizability of the ions in a dielectric medium in a self-consistent manner, etc. Addressing any of these aspects is certainly beyond the scope of the present work. Cognizant of the fact that specific details of ion-pairing depends on the identity of the ion, we sought out whether the charge symmetry breaking in PSBMA is universal or this is true only for K^+ . Indeed, as shown in the manuscript, charge symmetry breaking is observed for Li^+ as well. We have added a brief comment on this issue on pages 19 and 22.

10) Do the authors have any information on the ion-charged group interaction on the molecular level e.g. by MD simulation?

Response 1.10: We agree that it would be very helpful to gain insights at atomic/ionic level using techniques such as DFT and MD. Efforts in this direction are relegated to future work.

Overall, we are grateful to the reviewer for such a thorough reading of the manuscript and for several inputs to improve the clarity of the manuscript.

Response to Reviewer #2

We thank the reviewer for several useful comments to enhance the clarity of the presentation. We have addressed all points raised by the reviewer as detailed below and incorporated in the revised version and SI. The comments of the reviewer are in black and our response is in blue.

This work investigates translocation of polyelectrolytes through solid state nanopores. There are some interesting observations, such as the reversal of direction when the chemical structure is reversed. However, the qualitative explanation for the observed phenomenon lacks independent validation and the quantitative predictions have too many uncertainties. Additionally, even if revisions are made I think this work could be more suitable for a journal dedicated to macromolecules, such as ACS Macromolecules.

Response 2.1: We respectfully disagree with the reviewer's opinion regarding the suitability of the present manuscript for *Nature Communications*. As mentioned by the other two reviewers, the results presented in the manuscript are significant and of broad interest. The present work opens a new avenue of research in the uncharted territory of a large class of biological and synthetic macromolecules composed of zwitterionic (dipolar) moieties. This Communication reports the first experimental investigation of such systems. This work is the tip of an iceberg and is expected to generate active future research in the research community as indicated by the curiosity, excitement, and endorsement expressed by the other two reviewers' reports. The reported results are significant and novel, with potential impact on fundamental understanding of a large class of macromolecules not yet investigated, and with implications in many applications such as protein separation. In view of these features, the present work is ideally suited for the broad readership of *Nature Communications*.

The main claim of this work is that polyelectrolytes exhibit electrophoretic mobility. However, only resistive pulse sensing with nanopores is used for investigation. It is well known that the translocation through nanopores is not governed only by electrophoresis, also electroosmosis and dielectrophoresis come into play: <https://dx.doi.org/10.1021/acsnano.0c06981> How can the authors ex-

clude these other effects?

Response 2.2: We agree that electroosmotic flow (EOF) is always present in nanopore experiments if the inner surface of the nanopore bears a certain charge density. We are fully aware of the role of EOF in nanopore experiments based on our own research on this issue for more than a decade (Refs. 18, 41, and 47) and several other works in the literature. We should have mentioned the role of EOF in the original version, but now we have addressed EOF in the revision. While the research community is aware of the potential role played by EOF, to-date, there is no direct experimental measurement of EOF in nanopore experiments. However, our experimental results clearly show that the electrophoretic flow dominates over the EOF. As a specific example in our system, PSBMA moves towards the positive electrode by overwhelming the EOF (arising from the negatively charged pore wall) that works in the opposite direction. We have now included a discussion of EOF in the revised version on pages 4, 7-8, 13, and 16. We thank this reviewer (and the third reviewer) for their suggestion to include a discussion on the role of EOF.

On the other hand, there is no dielectrophoresis in our system. In fact, in the quoted reference by the reviewer with the title containing dielectrophoresis, there is no dielectrophoresis at all, but only dipole-phoresis. The dipole-phoresis describing movement of entities with permanent dipole moment is fundamentally different from dielectrophoresis describing the movement of neutral dielectric entities capable of generating induced dipoles under an electric field. The force \mathbf{f} on a dipole of dipole moment \mathbf{p} in an electric field \mathbf{E} is $\mathbf{f} = (\mathbf{p} \cdot \nabla)\mathbf{E}$. For a permanent dipole, the force is linear with \mathbf{E} , as in the present paper. On the other hand, if the dipole arises as an induced dipole, as in the case of dielectrics, then the induced dipole moment is $\alpha v \mathbf{E}$ where α is the polarizability of the dielectric body and v is the volume of the body. Substituting this in the expression for the force, one gets $\mathbf{f} = (\alpha v \mathbf{E} \cdot \nabla)\mathbf{E} = \alpha v \nabla |\mathbf{E}|^2$. This is the force in dielectrophoresis (Ref.59). While the dipole-phoresis depends linearly on the electric field, the dielectrophoresis depends on the **square** of the electric field. To quote the discoverer (Herbert A. Pohl) of dielectrophoresis, from page 17 of his book on “Dielectrophoresis” (Cambridge University Press, 1978), “We may summarize the force and the energy depend upon the *square*

of the electric field intensity. The latter result emphasizes that dielectrophoresis is independent of the sign of the field direction. It can occur in an a.c. field.” By simultaneously altering the signs of the permanent dipole moment and electric field will give insensitivity of the force to the direction of the electric field, but this force is still linearly proportional to \mathbf{E} and not as in the case of dielectrophoresis. Obviously, electrophoretic mobility of our molecule towards its favored electrode (and not in the other direction) is not dielectrophoresis. We have now added a brief discussion about dielectrophoresis on page 4.

There is a lack of complementary techniques to verify the electrophoretic mobility although it is indicated by the linear relation of voltage and event rate. It would be very simple to test mobility under a field in bulk solutions. The molecules can be labelled if need be. In fact, it is hard to understand why the nanopore system was chosen to begin with if the purpose is to investigate electrophoretic mobility.

Response 2.3: Our purpose is not just to investigate electrophoretic mobility. One of the grand challenges in the physics of biomacromolecules and synthetic charged macromolecules is to achieve a fundamental understanding of the dielectric function in the neighborhood of ionic species in crowded conditions. This understanding is vital for all aspects of structure, dynamics, and functions of these systems and specifically for the effective charge of the ionic species constituting the macromolecules.

Of course, it would be desirable to have other independent techniques to investigate the local dielectric function in the immediate neighborhood of the backbone of flexible polymer molecules and whether local charge neutrality (within a local distance of ~ 1 nm) is insensitive to local dielectric constant. But, we are not aware of any such techniques. This is why we designed the experiment using single-molecule electrophoresis through a single nanopore to address our targeted questions. Our experiments then revealed the unexpected results reported here. The rationale for our experimental design, instead of any bulk measurements, is as follows.

In bulk solutions, a dispersed flexible polymer such as PSBMA or PMPC is not a rigid sphere, a rigid cylinder, or a rigid ellipsoid. The molecule adopts enormous number of conformations and there is a perpetual change in its conformations. Concomitant with these conformational fluctuations, the dipolar orientations

of the zwitterionic repeat units (and there are hundreds of such repeat units in one molecule) are constantly changing. The net outcome is that these orientations and chain conformations are averaged out resulting in a coil-like statistical fractal object. Such a conformational status of the molecule prevents the distinction between the ionic groups on the repeat unit in terms of their normal distance from the chain backbone. This is also true with zeta potential measurements as well as the standard gel electrophoresis (with the typical large and heterogeneous mesh sizes). Therefore, we designed our experiment by trying to transport the molecule through a nanopore as an unfolded single file. In this manner, the ionic groups constituting every repeat unit are exposed locally in contrast with the situation of heavily interpenetrating chain segments inside a coil.

We have now added a discussion of the motivation for our experimental design on pages 2-4.

I also think that even for the nanopore measurements alone, the study could be extended to see what happens at lower ionic strength or different pH, and by using zwitterions with weak acid or base groups. These are quite obviously important parameters to vary and would shed light on the mechanisms. Complementary experiments on polyelectrolytes would also be very useful.

Response 2.3: We agree that further exploration by varying ionic strength, pH, and chemistry would provide more results. While these parameters are trivially changed in simulations, there are certain practical challenges in experiments. For example, if the ionic strength is low, the measured pore current can be so low that accurate data acquisition of signatures of the interference between the polymer and the pore becomes difficult. Nevertheless, we plan to strategically pursue the effect of these variable in the future.

As far as I understand the main physical explanation for the findings is related to a varying degree of counterion condensation on the outer vs inner groups. This is reasonably well described in the “Charge regulation by counterion binding” section. However, some wording is still misleading. Why is “degree of ionization” discussed? Both the negative and positive charges are strong and always ionized. I also think the naming “charge symmetry breaking” is confusing. The distribution

is still cylindrically symmetric?

Response 2.4: We thank the reviewer for this helpful comment. In general for charged macromolecules, both the degree of ionization of a native ionic monomer (obtained from titration curves for example) and the **effective** degree of ionization of that monomer as a part of a macromolecule (after accounting for counterion condensation) are generically called ‘degree of ionization’. In view of the reviewer’s comment, we have now mentioned that the “charge regulation by counterion binding” leads to an ‘effective degree of ionization’ on pages 2-3 and 19. Regarding CSB, we mean that in the state of CSB, the dipolar unit carries a net charge (due to varying degree of counterion condensation) in contrast to the situation of net charge being zero for the dipolar unit in the absence of any counterion condensation. In our theoretical model, we assume cylindrical symmetry about the chain backbone axis. We have added a few words on page 6 and in Supplementary Note 3 regarding the above comments.

The theoretical model of translocation time is quite detailed but a bit cumbersome to follow for a non-specialist. I believe it could be shortened in the main text and extended in a supporting information section. More importantly, there are many unknown parameters in this model. It appears that the purpose is to get an independent estimate of translocation time to be compared with experiments so that the charge of the translocating species can be obtained. However, my impression is that there are too many uncertainties or vaguely motivated values (e.g. interaction energy with pore wall) to trust the outcomes quantitatively. In particular, a specific value of the degree of counterion condensation on the outer groups is assumed (0.25) and this seems rather arbitrarily, yet it is critical for obtaining the degree of counterion condensation on the inner groups.

Response 2.5: Construction of an adequate, but even the simplest, theoretical model to capture the essential physics of the rich experimental system investigated here inevitably requires treatment of many details. These include the size and orientation of the zwitterionic repeat unit with respect to the chain backbone, distance between the ionic groups and dipole moment of the zwitterionic unit, separation distance between two adjacent repeat units, thickness of the backbone, chain length of the

polymer, pore length, pore diameter, surface charge density on the inner wall of the pore, ionic strength, pH, identity of counterion, local dielectric constant, and externally imposed electric field profile across the nanopore. Naturally, such a rich system needs parametrization of the various contributing factors. In fact, contrary to the reviewer's impression, we have taken reasonable values of these parameters in our theoretical predictions, as clearly explained in the original version. The choice of the values of these parameters is neither vague nor arbitrary, but was driven by physicochemical details obtained from the literature and molecular mechanics simulations. The value of 0.25 for the effective charge of the outer group of the zwitterion comes from the experimental results in the literature (Ref.17), as already mentioned in the original version. We would like to emphasize that the key result is not its specific value, but the difference in charges between the inner and outer groups.

In addition to providing effective charges of the ionic groups, the most important merit of the presented theoretical model is that it provides conceptual insights into the relative contributions of conformational entropy, electrophoretic drift-diffusion, and pore-polymer interaction. For example, using reasonable values of the parameters from the literature, we can rationalize why the translocation time depends exponentially on voltage and not inversely on the voltage that has been seen in many nanopore experiments. We find the exponential dependence to originate from the loss of conformational entropy of the polymer as it threads through the nanopore.

We have included a discussion of the above comments in the revised version on pages 16-18. In addition, as suggested by the reviewer, we have now moved some parts of the theory section (model, equations 2 & 5-8 and Tables I and II) to Supplementary Notes 3-6 and Supplementary Tables 1-2.

Shape asymmetry may also influence translocation directionality. Do the pores exhibit rectification? The study should include characteristic IV curves and as a control experiment, it should be verified that translocations occur (or do not occur) when the voltage is reversed over the same pore and molecules introduced on the other side.

Response 2.6: The pores do not exhibit rectification. We worked with the pores

only if they do not exhibit rectification as mentioned on page 24. An example of the I-V curve is given as Supplementary Figure 14 in Supplementary Information. Also, we do not observe any shape asymmetry effect. Translocations of PSBMA occur when the voltage is reversed over the same pore and the molecule is introduced on the other side as shown in Fig.2 in this response.

FIG. 2: Pore geometry independence of CSB of PSBMA. 50 nM PSBMA in 3.6 M LiCl buffered at pH 6 with 10 mM HEPES was used. After recording current traces with PSBMA in one side of the silicon nitride membrane in both voltage polarities (a), the flow cell was disassembled, and the membrane was flushed with Milli Q water. Then, the flow cell was reassembled, and another measurement was done with PSBMA on the other side of the membrane (b). Pore diameter: 4 nm, voltage: 300 mV, sampling rate: 250 kHz, low-pass filter frequency: 10 kHz.

I cannot see how the folded events are separated from the unfolded ones. Their distributions are clearly overlapping (Fig. 3b). Also it seems that unfolded events may be missed based on the distribution of dwell times (Fig. 3d). What is meant by “most of those events show two different blockage levels” in the quantitative sense?

Response 2.7: Yes. The two distributions are overlapping, and thus, we have taken only half of each distribution in our data analysis as we already described on pages 8 and 21 in our original manuscript (pages 12 and 25-26 in the revised manuscript).

We now have addressed quantitative analysis of different levels in deep blockages in Supplementary Note 1. Using OpenNanopore software package and our custom MATLAB code, we found at least 93% of deep blockages have two different blockage levels at all voltages, which is now mentioned on page 9 of our revised manuscript and discussed in Supplementary Note 1.

We thank for the helpful comments.

The threshold behavior in voltage is speculative. Why not perform measurements at lower voltages to properly investigate?

Response 2.8: Indeed, we did perform many experiments for voltages at lower and lower values. As well known in the experimental community working on single-molecule electrophoresis through single nanopores, getting statistically meaningful data at very low voltages is difficult. First, the noise in the pore current at low voltages can prohibitively lower the signal-to-noise ratio in the experiment. Second, the capture rate is very low at low voltages and it is difficult to collect enough data that are statistically significant in data processing, while maintaining the robustness of the pore diameter.

If N and M denote number of monomers, not length, dividing the voltage with these numbers does not give a field.

Response 2.9: As already spelled out in the original manuscript, these numbers are in the unit of monomer length. No change is made.

I believe the titration curves are not very interesting and can be moved to supplementary information.

Response 2.10: We disagree. The inclusion of the titration curve right away offers

the premise and perspective of the conceptual issue addressed in this manuscript. Also, please see the response to the first reviewer.

Fig. 3f should not be needed as it repeats information.

Response 2.11: We prefer to keep Fig.3f even though it is repetitious of the bottom inconspicuous curve in Fig.3e, so that the voltage dependence can be easily recognized by the readers.

The structure of the paper is strange because central results are presented already before the results section.

Response 2.12: We prefer to keep the original style of presentation, because we believe that this style is likely to be preferred by most of the readers.

Both the introduction and discussion sections contain strange comments on the connection to biological system. I do not see the relevance since transport in cells does not occur by electric fields.

Response 2.13: We are surprised by these comments. In fact, all biological systems are composed, sustained, and functional as a Coulomb soup, with regulated modulations in local electric fields and local dielectric heterogeneity. This is copiously evident in any introductory textbook on Neuroscience, for example.

Page 19: “the local translocation speed of the monomers of the polyzwitterion is essentially uniform at all voltages in the present study” Please clarify, if the dwell times vary the speed of the monomers must also vary?

Response 2.14: We agree with the reviewer and we thank the reviewer for pointing this out. On page 20, we have now spelled out that we assume that all monomers of the chain in each translocation event have uniform velocity, without any additional nonlinear effects such as tension propagation forces seen in other systems as in Ref.40.

Page 20: “wavevector dependence of the dielectric constant” What waves?

Response 2.15: We are stunned by the reviewer’s question, unless we have misunderstood the question. The so called dielectric constant of a dielectric medium is not a constant, but is a function of spatial location and time. In all theories of dielectrics, this dielectric function is expressed in terms of wavevector and frequency which are

Fourier conjugates to spatial location and time, respectively. Indeed, we are talking about electromagnetic waves in dielectrics (which are responsible for phenomena such as dielectrophoresis and intermolecular van der Waals attractions).

Overall, we would like to thank the reviewer for the investment of time and care, and for several useful comments to enhance the clarity of the presentation.

Response to Reviewer #3

We thank the reviewer for appreciating the importance of the study and for offering excellent suggestions to enhance the manuscript. We have positively addressed all points raised by the reviewer as detailed below and incorporated in the revised version and SI. The comments of the reviewer are in black and our response is in blue.

The manuscript by Lee and Muthukumar studies the important topic of the translocation characteristics of polar and neutral polymers through nanoscale pores. There has been a lot of work on understanding the passage of uniformly charged polymers through nanopores for DNA sequencing applications, but much less work on so called zwitterionic polymers. This is of growing interest with the emergence of new applications nanopores for proteomics and glycomics. The authors proceed with studying two simple polymers [poly(sulfobetaine methacrylate) (PSBMA) and poly(2-methacryloyloxyethyl phosphorylcholine) (PMPC)] as model systems for the behaviour of zwitterionic polymers. This type of work is important in order to develop a better understanding of their passage dynamics and thus detection strategies for improved sensing performance. The authors find that the charge of the zwitterionic group farthest from the backbone dominates the electrophoretic response of the polymer. This is attributed to less shielding of this ionic group of the zwitterionic monomer. The authors refer to this differential counterion binding phenomenon as a charge symmetry breaking (CSB).

Response 3.1: We thank the reviewer for such an excellent succinct summary of our results and their potential broader impacts.

Although the topic and ideas introduced in the manuscript are of high interest, the results presented in the manuscript in its current form do not fully support the conclusion. More experimental work and/or analysis is needed before it can be considered for publication. At the moment, the interpretation of the observed nanopore signals, and the experiments shown do not sufficiently support charge symmetry breaking as being the phenomenon responsible for the passage characteristics.

Response 3.2: We thank the reviewer for offering constructive comments to satisfy the reviewer in order to fully validate the proposed mechanism of CSB. Stimulated by

this reviewer's comments, we have performed additional experiments and analysis. We believe that the results from this additional work successfully validate the main theme of CSB presented in the original manuscript. Our new results and detailed responses are given below along with the specific suggestions from the reviewer.

Below is a list of major points that need to be addressed:

- The manuscript would benefit greatly from a clearer physicochemical description of the polymers used in its study, i.e. polymer length, cross-sectional diameter or persistence length. These parameters are crucial for helping readers interpret nanopore translocations signals and relating them to the conformations held during passage. Namely, Figures 3b and 3c show single-file blockages centered around $I_b/I_o \approx 0.6$ which, to first order, suggests a polymer cross-section diameter of $d \approx d_{\text{pore}} \sqrt{(\Delta I/I)} \approx 2.5 \text{ nm}$. Is this the value expected for a elongated polymer, or does this indicate that polymers are more blob-like inside pores? Given that polymers being stretched inside pores is a central assumption of the theoretical model, experimental data should be used to better interpret polymer conformations during translocation. For example, ssDNA does not produce quantized blockages unlike dsDNA because of their very different persistence length. Why are these polymers expected to translocate as single-files with some folds and not as blobs?

Response 3.3: We agree that we should have already provided the physicochemical characteristics of the polymer. We have now inserted these details in the revised version on pages 7 and 16. These are as follows. The nominal degree of polymerization of PSBMA sample is 513, and with the length of repeat unit of about 0.25 nm, the average chain length is about 0.128 μm . Only average values of these measures are available for our synthetic model compounds, in contrast with biological samples such as polynucleotides and proteins. Based on molecular mechanics simulations, we have determined the cross-sectional diameter of PSBMA as 2.2 nm. From our light scattering and Langevin Dynamics simulation studies, we are aware that the PSBMA and PMPC polymers are self-associating into essentially globule-like structures in salt-free dilute solutions due to inter-monomer dipole-dipole attractions. However, as the salt concentration is increased to a level of 1 M KCl (pertinent to the present experimental conditions), the chain swells into flexible coil-like confor-

mations exhibiting the so-called ‘anti-polyelectrolyte effect’ arising from the electrostatic screening of dipole-dipole interactions. Thus, at the experimental conditions in the present work, both polyzwitterions (PSBMA and PMPC) belong to the category of flexible polymers. Even though we could not directly measure the persistence length of these molecules, we believe that it is about 2 nm, based on our experience with flexible polymers of similar chemical composition. Therefore, assuming that the chain obeys Gaussian coil statistics, the radius of gyration R_g in solutions is 9.2 nm. This value is actually a lower bound, because the chain is expected to swell beyond its Gaussian statistics due to excluded volume interactions.

As a summary of the above arguments, for PSBMA in our experiments, the cross-sectional diameter of the polymer is 2.2 nm, chain length is 128 nm, and radius of gyration in solution is 9.2 nm. As astutely pointed out by the reviewer, the cross-sectional diameter that can be qualitatively inferred from the ratio of blocked current in single-file blockages to the open pore current is ~ 2.9 nm, consistent with the value we obtained from molecular mechanics. Further, R_g is more than twice bigger than the pore diameter. Therefore, there is no possibility for the chain to translocate as either as a single blob or a sequence of blobs which would be unfavored due to conformational entropic penalty. In view of these considerations, we identified the mode of translocation as threading of extended conformations as in the well-established nanopore-based sequencing of polynucleotides and proteins. Thanks to the reviewer, we have now presented these important details on pages 7 and 16 in the revised version.

- The interpretation of single-file and folded translocations should be elaborated more or simply revisited. Namely, are events interpreted as folded actually folded or correspond instead to two polymers inside the pore at once? The latter seems plausible given the similar timescale of dwell times and inter-event times: Figure 3 shows that at 150mV $\langle\tau\rangle\approx 7.5$ ms and $\tau_{-a}=1/60Hz=17$ ms. Do folded translocations always begin with a deeper blockade, as expected from polymers bending inside the pore? This should also be clarified.

Response 3.4: We agree with the reviewer’s suggestion to elaborate on the polymer conformations responsible for the deep blockades.

We have addressed the question of whether folded translocations always begin with a deeper blockade, by performing detailed analysis of different blocked current levels and their sequences in the various folded translocation events. The answer is ‘no’.

The above conclusion is based on the following procedure. Using the OpenNanopore software to get amplitude and duration of each level in multi-level events, we plotted the accumulated duration *versus* I_b/I_0 at each voltage by using our custom MATLAB code. Aside from the absence of a detectable double-folded population and an increase in the fraction of moderate blockages, these plots show similar distributions as I_b/I_0 histograms given in Figure 3c and Supplementary Figure 3, which were constructed using the MATLAB code written by Plesa and Dekker (Ref. 35) and OriginPro (OriginLab Corporation, USA). An example (120 mV) is given in Supplementary Figure 17. For further analysis, we grouped the current level in the deepest I_b/I_0 as level 2, the moderate level as level 1, and the shallowest level as level 0. Using our custom MATLAB code, we found that only 0.82 to 6.6% of deep blockages (events containing level 2) were single-level events, and 59 to 96% of deep blockages have more than 2 steps. Among four possibilities of combination of the first level group and the last level group (1/1, 2/1, 1/2, 2/2) for multi-level deep events, the case starting with moderate blockage step and ending with moderate blockage step (1/1) is the most probable at all voltages ranging from 42 to 82%. This preferred sequence of current levels in deep blockages supports our ‘chain-folding’ scenario since it is probable that unfolded part of chain is inserted to the pore entrance first and then chain folds and unfolds within the pore during the translocation until unfolded tail of chain finally goes through the narrow pore.

Regarding the discussion of ‘folded or two polymers’ responsible for deeper blockages, we believe that it is unlikely that two molecules simultaneously move through the nanopore. However, there are two possible explanations for the deep blockades, given the cross-sectional diameter of the polymer in the range of 2.2-2.5 nm and the pore diameter of 3.7 nm. In the first explanation, the chain conformation inside the nanopore can adopt a short hairpin-like kink which then proceeds to complete the translocation process. The formation of such kinks is not prohibitive, because

the persistence length (which is an ensemble average) of the polymer is comparable to the pore radius and there are always conformational fluctuations at the experimental conditions. The kink formation does not necessarily have to occur from the start of the pore entrance but can occur at any location inside the nanopore. This is consistent with the experimental observation that the folded translocations do not always begin with a deeper blockade. Another possible explanation is that a second molecule can try to enter the pore during the sojourn of the first molecule inside the pore. In this scenario, deeper blockades will occur at random times depending on the arrival frequency of the second molecule as well as the lingering time of the second molecule not being able to penetrate into the pore because there is already the first molecule inside the pore. While the voltage dependence of fraction of deeper events in Supplementary Figure 2 can be rationalized with this scenario, this cannot explain the duration of the deeper blockade events decreasing with voltage as in Figure 3f and 4d. Indeed, more insight can be gleaned by performing detailed Molecular Dynamics simulations with explicit water, ions, and chemical structure. We plan to perform such non-trivial simulations in the future. We have now provided the above explanations on page 10 in the revised manuscript and in Supplementary Note 1. Thanks, indeed.

- It should be further noted that an exponential dependence of translocation time on voltage is not common for electrophoretically-driven translocations which instead commonly result in inverse voltage relationships: $\tau \sim \Delta V^{-1}$. Exponential relationships have instead been observed when a molecule needs to overcome a strong energy barrier to enter pores, such as for DNA hairpins through biopores or translocations of long self-hybridized ssDNA through solid-state nanopores.

Response 3.5: We thank the reviewer for urging us to point out the important implication of our observation. Yes, the observed exponential dependence of the translocation time on the voltage is due to a free energy barrier. This observation is consistent with the predictions from our theoretical model, and we find the major contributor to the free energy barrier arises from the loss of conformational entropy during translocation. We have now elaborated this point on page 12 in the revised manuscript.

- Strikingly, the argument that PMPC translocates towards the negative electrode as a polyanion is not a convincing proof of Charge Symmetry Breaking, given that electroosmotic flow (EOF) arising from negatively charged SiN also flows towards the negative electrode. Additionally, the section addressing PMPC translocations would highly benefit from a more in-depth analysis like the ones shown for PSBMA, i.e. capture rate and translocation time vs voltage. In addition to EOF surprisingly never being discussed as a source of variation for the CSB phenomena, other important physical phenomena were dismissed. For instance, the electric field inside a nanopore is non-uniform, and is strongest towards the pore walls, i.e. towards the ionic groups further away from a stretched-out chain backbone.

Response 3.6: We thank the reviewer for these significant comments pertinent to the core of our work. We have addressed these comments in detail by performing new experiments and calculations. All of these new results provide convincing proof of our CSB mechanism presented in the original manuscript.

(1) At pH 7, the silicon nitride pore is negatively charged. In this condition, the EOF is in the direction towards the negative electrode. Yet, PSBMA moves towards the positive electrode, behaving like a polyanion. Therefore, it is evident that the electrophoretic force on the polymer overwhelms the opposing EOF.

(2) Under the same experimental conditions as in (1), we find that PMPC moves towards the negative electrode, behaving like a polycation. We argued that this polycationic behavior is due to CSB analogous to the phenomenon for PSBMA. However, as the reviewer has correctly raised, this is not a direct evidence for CSB in PMPC, because both the electrophoretic force and EOF are in the same direction. In order to determine whether EOF is a minor contributor or not, we have designed new experiments as detailed in Response 3.9. We created EOF in the opposite direction to the above case (1) by making the nanopore surface to bear positive charge. This is accomplished by performing the experiments at pH 3.7 (so that the pore is positively charged and the zwitterion ionic moiety is net charge neutral in the absence of CSB that arises from counterion binding). Please see Response 3.9 for more details. Under these conditions, EOF is in the direction towards the positive electrode. Yet, PMPC moved towards only the negative electrode and not at all in

the direction of the EOF. Considering the importance of this discussion, we have now replaced the earlier data in Figure 4 (Figure 5 in the revised manuscript) by the data in the new experimental condition at pH 3.7. We have now addressed these discussions (1) and (2) in the revised version on pages 4, 7-8, 13, 16.

(3) We have tried to systematically study the voltage dependence of the translocation time for PMPC. In our experiments with PMPC, the translocation is quite fast primarily due to the polymer length being not too long (degree of polymerization is only ~ 100). However, a few preliminary experiments show that the translocation time decreases as the voltage is increased. We do not yet have enough statistics to publicly present systematically investigated results. However, the primary purpose of even considering PMPC in our manuscript is to show the altered direction of movement upon a reversal of the polarity in the zwitterion, in support of the proposed CSB mechanism. Since this goal is achieved, we relegate systematic investigations of PMPC and other polyzwitterions to future work. This issue is now addressed on page 16 in the revised manuscript.

(4) Regarding the nonuniformity of the electric field inside the nanopore, we have computed the three-dimensional electric field profile resolved into the component in the direction of the external field and the radial component (by assuming cylindrical symmetry, which is a good approximation for solid-state nanopores). Of course, as expected, the radial component of the electric field is the strongest at the pore wall, satisfying the contact theorem, and it decays quickly in the radial direction towards the central axis of the pore. The important component is that in the direction of translocation, and it is essentially uniform within only 2% change as the radial distance from the central axis is changed from zero to 1.09 nm. Thus we do not believe that the observed CSB can be attributed to inhomogeneity of the electric field in the present system. In addition, we found that the difference of electric field in radial direction does not change with voltage, which cannot explain $\langle \tau \rangle$ decreasing with voltage. We have now added this discussion in the revised version on pages 7-8.

- The content of this research is very nanopore-centric which, as discussed, can introduce forces and phenomena specific to nanopores, as opposed to the more gen-

eral electrophoretic mobility of neutral polyzwitterions, as suggested by the title. This research and its conclusions could also benefit from running gel electrophoresis experiments to support its conclusions and interpretations.

Response 3.7: Yes, this is nanopore-centric. However, this is on purpose for the following reasons.

As responded to the second reviewer, our purpose is not just to investigate electrophoretic mobility. One of the grand challenges in the physics of biomacromolecules and synthetic charged macromolecules is to achieve a fundamental understanding of the dielectric function in the neighborhood of ionic species in crowded conditions. This understanding is vital for all aspects of structure, dynamics, and functions of these systems and specifically for the effective charge of the ionic species constituting the macromolecules.

Of course, it would be desirable to have other non-nanopore-centric techniques to investigate the local dielectric function in the immediate neighborhood of the backbone of flexible polymer molecules and whether local charge neutrality (within a local distance of ~ 1 nm) is insensitive to local dielectric constant. But, we are not aware of any such techniques. This is why we designed the experiment using single-molecule electrophoresis through a single nanopore to address our targeted questions.

In general, in bulk solutions, a dispersed flexible polymer such as PSBMA or PMPC adopts enormous number of conformations and there is a perpetual change in its conformations. Concomitant with these conformational fluctuations, the dipolar orientations of the zwitterionic repeat units (and there are hundreds of such repeat units in one molecule) are constantly changing. The net outcome is that these orientations and chain conformations are averaged out resulting in a coil-like statistical fractal object. Such a conformational status of the molecule prevents the distinction between the ionic groups on the repeat unit in terms of their normal distance from the chain backbone. This is also true with zeta potential measurements as well as the standard gel electrophoresis (with the typical large and heterogeneous mesh sizes). In the particular case of gel electrophoresis, if we could synthesize gels with nanoscopic mesh sizes, there is a chance of reaching our goal. The ultimate limit

of such small meshes is the single nanopore system. Therefore, we designed our experiment by trying to transport the molecule through a nanopore as a single file. In this manner, the ionic groups constituting every repeat unit are exposed locally in contrast with the situation of heavily interpenetrating chain segments inside a coil.

We have now added a discussion of the motivation for our experimental design on pages 2-4.

- The authors write on page 2: “even the simplest limit of homopolymers made of only neutral groups (so that complications from sequence effects can be deliberately suppressed) is yet to be investigated.” But a lot of work has been done studying the passage of PEG of different molecule weight through nanopores that are neutral or very weakly charged as well as proteins and peptides so that statement should be revised.

Response 3.8: We agree. We have now clarified the difference between the genuine neutral polymers such as PEG and the polyzwitterions made of “neutral zwitterions”, thanks to the first reviewer’s comments. In our discussion of this aspect, we have now included a few sentences on the single-molecule electrophoresis studies on PEG and other neutral polymers on page 2.

- Performing more experiments while changing the pH of the solution would really provide more results to better understanding the regime of passage (CSB dominated or not), by changing the protonation state of the polymer and also of the pore wall (affecting EOF).

Response 3.9: We agree with this excellent suggestion from the review to perform additional measurements by changing pH. Accordingly, we have performed the following new set of experiments. The aim is to unambiguously buttress our claim that the observed CSB for PMPC is electrophoretically driven and not by EOF. In order to achieve this goal, we need to identify an experimental condition satisfying two criteria: (a) the nanopore surface is positively charged, which would create an EOF in the direction towards the positive electrode, and (b) the protonation of the ionic groups is such that the net charge of the zwitterionic group is neutral. Under these conditions, if we observe mobility of PMPC in the opposite direction to that of EOF,

then we can clearly conclude that the EOF is not the contributing factor for the observed CSB for PMPC. In order to meet the above two criteria, we have chosen pH 3.7 for the translocation experiments. The rationale for this choice is as follows. To reach the appropriate pH condition to ensure that the pore wall is positively charged, we made conductance measurements with our silicon nitride nanopores at different pH values. As shown in the illustrative example (Supplementary Figure 1), the pore conductance is a minimum at the isoelectric pH of about 6 according to a previous work (Ref.64). Therefore, we investigated the translocation of PMPC at pH 3.7 (below the isoelectric pH value) enabling EOF in the direction towards the positive electrode. Furthermore, at pH 3.7, which is higher than pKa (~ 1.0) of polyphosphoric acid by more than 2, the protonation state of PMPC would have only neutral MPC units as at pH 7. With this design of the experimental condition, PMPC in the positive configuration (Figure 5d) moved only towards the negative electrode where the contribution from the EOF is obviously overwhelmed by the electrophoretic force. Of course, in the negative configuration (Figure 5c), no translocation event was observed. Thus the observed CSB cannot be attributed to EOF.

Other minor comments

- Each captions should include the experimental parameters used to acquire the results, pore size, voltage, concentration of analyte, of salt, sampling rate, low-pass filter value, etc. . .

Response 3.10: We thank the reviewer for this helpful comment. We added the experimental parameters pointed by the reviewer in the captions of Figures 1-5 and Supplementary Figures 2-10 and 15-16.

- Capture rate data should be reported in Hz/nM.

Response 3.11: We appreciate the reviewer for pointing out this. We re-plotted the capture rate data in Figures 3(a) and 4(c) with y-axis in Hz/nM. We also changed the slope value of R_c vs. voltage plot on page 10 into 4.7×10^{-3} in Hz/nM · mV.

- With a >40 Hz capture rate, why show plots with only <4000 events, this represents only 100 seconds of recording. It is statistically enough, but nanopore experiments are often carried out for longer.

Response 3.12: Yes. The measurement could have been carried out for much longer durations as long as the pore size is stable. Contrary to typical protein pores, silicon nitride pores are known to increase (or decrease) their sizes over time depending on the initial pore size, pH, electrolyte identity, etc. [Li, Q. et al. Size evolution and surface characterization of solid-state nanopores in different aqueous solutions. *Nanoscale* **4**, 1572-1576 (2012); Yanagi, I., Akahori, R., & Takeda, K.-I. Stable fabrication of a large nanopore by controlled dielectric breakdown in a high-pH solution for the detection of various-sized molecules. *Sci. Rep.* **9**, 13143 (2019); Chou, Y.-C., Das, P. M., Monos, D. S., & Drndić, M. Lifetime and stability of silicon nitride nanopores and nanopore arrays for ionic measurements. *ACS Nano* **14**, 6715-6728 (2020).] In our experimental conditions, we found pore sizes to increase over time especially during measurement (translocation of polyzwitterions). Since one goal of our work is quantitative analysis of voltage dependence of τ , it was important to make sure the pore size has not increased during measurement. Thus, we made three strategies. 1. We recorded current traces starting at the highest voltage of the voltages we used and went to lower voltages so that we can make sure the decrease of τ and the increase of R_c with voltage is not caused by pore size expansion over time. 2. We spent only 1-2 minutes at 150, 175, 200 mV to minimize the measurement time, which minimized pore size expansion. 3. We measured the pore diameters again after the translocation measurements were done, and only data with stable pore sizes were used. We have added a discussion of this point on page 25. We thank the reviewer to help improve this manuscript.

- Why is the number of events changing in the various panels of figure 3?

Response 3.13: We understand the different numbers could be confusing. As we noted in the METHODS section on pages 25-26, n in Figure 3b is the number of events used in the scatter plot, and n in Figure 3c is the number of events used for double Gaussian fitting in the I_b/I_0 plot. n in Figure 3c is smaller than n in Figure 3b since we excluded unsuccessful translocations with $I_b/I_0 > 0.6$ for the Gaussian fitting. n_{folded} and $n_{unfolded}$ in Figure 3d are the numbers of events used for each log-normal fitting. As described on pages 12 and 25-26, only half distributions of each Gaussian distribution were used to avoid the overlap between the two Gaussian

curves, which made the sum of two n values smaller than n in Figure 3c. The meaning of different n values was made more clear in Figure 3 caption.

- Which experimental data is the model compared to in Figure 7? The authors should also compare with the LiCl data, which is known to shield more.

Response 3.14: We thank the reviewer for pointing out these. We added "The experimental data is from Figure 3(f)." in the legend of Figure 7.

For the LiCl data, both the experimental and theoretical values of $\langle\tau\rangle/\langle\tau\rangle_{300mV}$ exhibit the same common trend of $\langle\tau\rangle/\langle\tau\rangle_{300mV}$ decreasing with voltage, as shown in Supplementary Figure 11. However, experimental values of $\langle\tau\rangle/\langle\tau\rangle_{300mV}$ decrease more rapidly with voltage compared to the theoretical values. Perfect quantitative fitting between the present theory and experiments cannot be expected due to the well-recognized 'special ion' effect of Li^+ arising from its high electron density (Ref: Robinson, R. A. & Stokes, R. H. *Electrolyte Solutions* (Dover, 2002)). The Fokker-Planck formalism used in the theory is not capable of describing the specific details of hydrated Li^+ ion at different voltages. A more fundamental theoretical treatment of specificity of counterions in nano-confinement is beyond the scope of the present goal and is relegated to future investigations. This is now mentioned on page 19.

Overall. we are sincerely grateful to the reviewer for such an outstanding and scholarly review which has helped us to bring forth the richness of our discovery.

Response to Reviewer #1

We thank the reviewer for the recommendation to publish the revised manuscript. The comments of the reviewer are in black and our response is in blue.

All comments have been fully addressed, to my opinion. I recommend to publish the manuscript in its revised present form.

Response: We thank the reviewer for the recommendation and their contribution in the first round of review with excellent suggestions to enhance the manuscript.

Response to Reviewer #2

We thank the reviewer for their time to review and raise curiosity-driven remarks. The comments of the reviewer are in black and our response is in blue.

While the revised version addresses some of my comments in a satisfactory manner, others remain unresolved in my opinion and the authors have chosen not to perform many of the suggested experiments and corrections. As I mentioned, there are certainly interesting aspects of this work but I still cannot recommend that it is published (at least not in Nature Communications) unless additional work is performed to properly understand the proposed mechanism.

Response 2.1: We respectfully disagree with the reviewer's subjective remark regarding the suitability of this manuscript for NCOMMS, based on the same justifications we provided earlier (please see Response 2.1 in the previous review). The proposed mechanism is clearly validated, without any doubt, by the various experiments already presented in the paper, as evident from the other reviewers' comments. There is no need for any additional critical experiment for the main goal of the paper. Of course, the present work is only the 'tip of an iceberg' on a new polymeric system and opens a new avenue for a broader study of the role of all kinds of experimental variables. We are confident that the present results will catalyze new research in the broader community working on molecular biophysics and synthetic polyelectrolytes, in addition to our own plans.

Starting with the good news, the authors have addressed my (and other reviewers) comments on the role of electroosmosis and dielectrophoresis good enough. I agree that the electroosmotic force will be opposing and that dielectrophoretic effects should at least not exhibit directionality. The complementary IV curve is also appreciated and several clarifications have been done.

Response 2.2: We thank the reviewer for the acknowledgement.

My main remaining concern is that the electrophoretic mobility of the zwitterions, which must be regarded as the most central result in this work, has still not been investigated by any complementary method. The authors claim that this would not be relevant since the polymer would not assume a linear conformation in bulk solution. I fully agree, but neither do they in the reservoirs outside the nanopores,

and yet they are attracted to the pore, apparently by electrophoresis. As I'm sure the authors are aware, the behavior they observe (frequency is linear with voltage) is established to be due because of drift motion (dominating over diffusion) within a certain distance (several microns I believe) away from the pore (Wanunu et al. Nat. Nanotech. 2010 and others). This means that the polymers seem to be moving in the field outside the pore, where they clearly must be random coils (as drawn by the authors). Therefore it should be possible to observe such motion also with conventional electrophoresis methods, capillary or gel, contrary to what the authors claim. Nanopore sensors are in the end very complex systems that are not designed to investigate electrophoretic mobility. We have other established tools for this. I emphasize that such measurements are important because either way they will show that the current interpretation is not correct, or at least incomplete: If mobility is observed, it means that the charge symmetry breaking is not sufficient to explain electrophoretic mobility (as it is based on a linear conformation). If mobility is not observed, there must be another explanation why the polymers are driven towards the pore.

Response 2.3: Of course, CSB is present even in bulk solution. Otherwise the neutral dipolar polyzwitterion would not move under an electric field to be captured by the nanopore. Thanks to the reviewer's remark, we now emphasize this more clearly in the revised version on page 3. We agree with the reviewer's comment that the capture of polyzwitterions by nanopores shows that CSB must exist even in coil-like conformations. Our statement is that we cannot investigate the pure CSB at the level of one zwitterionic moiety, that originates from the gradient in the local dielectric constant from the chain backbone to bulk water without interferences from intra-chain interactions, in bulk conditions. As already pointed out in the manuscript, the capture rate of PSBMA towards the positive electrode is linear with voltage whereas that in the opposite direction is zero at all voltages, as shown in Figure 1 (given below) for the convenience of the reviewer. The capture process is a manifestation of capillary electrophoresis (the dimension of the donor and receiver chambers is 1 cm³). There is no need to perform separate capillary electrophoresis experiments in the context of the present paper.

FIG. 1: PSBMA behaves as a polyanion even in the bulk of the donor chamber as evident from the linear relation between the capture rate and voltage in the negative configuration (positive electrode in the receiver chamber), while the capture rate is zero and independent of voltage in the positive configuration (negative electrode in the receiver chamber).

Related to the above, it is well-known from many studies using zeta potential measurements that the effective charge of macromolecules may be counterintuitive. For surface tethered polymers, negative charges have been observed on “neutral” polymers (example Zimmerman et al. Langmuir 2005, 21, 5108-5114) by streaming potential. These effects are likely related to chemical affinities between certain ionic species and organic groups. The authors have performed no such characterization. Measurements can of course also be done in solution phase by phase analysis light scattering. This also illustrates why measurements at different pH may be very informative as even non-ionizable polymers may alter their electrokinetic properties, but the authors have (still) not performed any such tests.

Response 2.4: It is well-known in advanced research laboratories that zeta-potential measurements on flexible polymers do not provide correct net charge (and sometimes even the correct sign of the net charge). This situation is primarily due to the assumption in interpreting experimental raw data that the porous and fluctuating flexible random coil is a colloidal particle with an effective surface charge. For

this reason, we did not perform zeta-potential measurements. More importantly, development of a valid theory of zeta-potential for a fluctuating polymer coil is required before performing any zeta-potential measurements, which is beyond the scope of the present work. Please note that we have already reported critical data by varying pH in one set of experiments. We have extensive experience with light scattering of polyelectrolytes which is not suitable to study single molecules due to copious intermolecular aggregation (the so-called slow mode) at the usual polymer concentrations required for light scattering. No change is made.

Regarding the model and values of different parameters, I still see several issues. The authors write that Supplementary Note 3 should explain why their choice of values is reasonable, but I find no such motivation in that section. The value of 0.25 for the effective charge of the outer groups of the zwitterion is apparently taken from a study on DNA, i.e. quite a different molecule. The interaction energy between monomer and wall is estimated under the assumption that only electrostatics contribute. I doubt this is the case and (again) it would be possible to perform experiments with other techniques to obtain more information on interactions between the zwitterions and silicon nitride surfaces. These points raise the question whether the agreement between theory and experiment in Fig. 7b is simply a result of too much freedom in multiparameter tuning. To be fair, this may not be a major flaw, but I think the authors need to be more open with the fact that the uncertainty is high with respect to the values they pick.

Response 2.5: Yes. We agree that we should mention that the values of the parameters in the theory are not *ab initio* but have uncertainties. Although we have already stated that our theory is only zeroth-order mean field theory due to the inevitable complexities of the system throughout our manuscript, we now have emphasized this by adding a sentence on page 23 in the revised manuscript.

For the rest, I have no more comments that require addressing by absolute necessity. I just leave some more thoughts here for the authors that they can consider (or ignore if they wish).

- I find it unlikely that most readers will prefer to see results and discussion in the introduction part of a paper (before the Results section).

- I still do not see any clear relevance to biological systems. Like the authors say, in a cell there are plenty of charged molecules in a “soup” - where would there be directional mobility in a field? (An exception would be the potential across certain membranes but I still do not see any relevance to zwitterions in such scenarios.)

- I’m sorry for “stunning” the authors by asking what wavevector they were referring to. I do know that the dielectric constant has a frequency dependence, but I still do not understand what the authors are trying to say. Are there electromagnetic waves coming into play in this work?

Response 2.6: We wish to ignore responding to the second and third items, but for the first item, we agree to move the ‘summarizing paragraph’ in the Introduction to Results, as also suggested by the third reviewer. We have now done this revision ensuring coherent flow of presentation.

Response to Reviewer #3

We thank the reviewer for the endorsement and appreciation of our additional effort to unambiguously demonstrate the validity of the discovery. The comments of the reviewer are in black and our response is in blue.

We want to thank the authors for their detailed and transparent responses to the comments from the different reviewers. They have performed new experiments and added new valuable discussions, which have resulted in a strengthened version of their manuscript. Of particular note, the new experiment recording PMPC in pH 3.7 solution now removes any doubt of EOF being the main mechanism responsible for driving capture and translocation of the polyzwitterion through the pore and thus strengthens the case for the charge symmetry breaking phenomena introduced.

Response 3.1: We are gratified by the reviewer's acknowledgment of the additional significant effort we made in the revision in terms of performing new experiments and better clarifying the fundamental concepts in this work. Thanks.

Nonetheless, we feel that two points should still be better addressed. The CSB mechanism is sound, and well described, but the manuscript would significantly benefit from having clearer presentation and interpretation of some aspects of the data:

Response 3.2: Thanks to the reviewer's comment, we now have addressed these points as given below and in the revised manuscript.

- Nature of the signals. As of now, it is not clear what the translocation signals correspond to, i.e. what conformations are held by the polymers as they pass through the pore in the single-file or folded states. This is an important point, especially because of the theoretical discussion and derivations at the end of the manuscript, and should be presented more clearly:

Response 3.3: Although it is desirable to make an one-to-one correspondence between the polymer conformation inside the nanopore and the ionic current trace, this is a difficult task even for polymers such as polystyrene sulfonate and DNA which have high charge density. The difficulty is even more severe for flexible chains with very weak charge density that are investigated in the present work. However,

we have already presented a few conjectures as plausible modes of conformations of the polymer inside the nanopore that are compatible with the observed ionic current traces. The population of events with intermediate blockade is assigned to single-file translocation with minor unstructured conformational fluctuations. The population of events with deeper blockades corresponds to translocation events where the polymer adopts conformations with one or more local short hernia-like structures inside the nanopore. Such structures can easily arise in the present system due to the flexibility of the chain as well as local association of dipolar zwitterionic monomers. Even in the intermediate blockade events, there ought to be some conformational fluctuations, albeit weak compared to those in deep blockade events, as mentioned above. Please see the cartoon given in Supplementary Figures 5 and 7 in the current revised version. As already mentioned in the earlier revised version, we do not know the precise conformations of our polymers inside the nanopore. We have again reiterated this in the current version. Nevertheless, based on the information given below in Response 3.4, our hypothesis on the conformations for deeper blockades seems reasonable. Furthermore, to avoid any confusion between such intra-pore structures with local kinks or nano-cilia (or ‘nano-blobs’) and the ‘folded’ conformations observed in DNA translocation, we have replaced the word ‘folded’ with ‘intra-pore-structured’ for the events with deeper blockades. Also, the theory presented in the manuscript is only for the single-file translocation and not for translocation modes with substantial conformational fluctuations consisting of local kinks. In comparing theory and experiments, we have considered only the data corresponding to the single-file mode. The weak conformational fluctuations in the single-file mode is incorporated into the effective friction coefficient k_0 per monomer in the theory. All of these explanations are now clearly presented on pages 11 and 12 in the revision.

- On page 16: “The cross-sectional diameter that can be qualitatively inferred from the ratio of blocked current in single-file blockages to the open pore current is ~ 2.9 nm, consistent with the value obtained from molecular dynamics.” How was the 2.9 nm value obtained? Although 2.9 nm is only 30% off from the expected 2.2 nm diameter, i.e. $(2.9-2.2)/2.2$, the current blockages which roughly scale with the squared diameter would be off by 74%: $(2.9^2-2.2^2)/2.2^2$. This significant deviation should be addressed.

Response 3.4: Please note that the estimated diameter (2.2 nm) was obtained only from its chemical structure without consideration of the hydration shell from water, as already mentioned in the earlier revision. It is known that polyelectrolytes strongly acquire water molecules around them making them as important materials for anti-fouling and other applications. Also, computer simulations (Clark, *et al.* J. Phys. Chem. B 2023, 127, 8185-8198; doi.org/10.1021/acs.jpcc.3c03654) have shown that water molecules are bound to polyelectrolyte backbone that can extend to multiple layers and that the mobility of water molecules near the backbone returns to the bulk solvent mobility only at about 1.8 nm from the backbone. Of course, most of the water molecules bound to the polymer are expected to be stripped away when the polymer undergoes translocation. However, as a conservative estimate, if we take the length of the pendant zwitterion group to be extended by a layer of only one bound water molecule (of length 0.28 nm), we estimate the effective diameter of the polymer as $2.2 + 2 \times 0.28 = 2.76$ nm (with an uncertainty of only $\sim 9\%$ in the cross-sectional area). Of course, this is not rigorous, but the estimates of what is expected from the hydrated chemical structure and the level of blocked current are reasonably close. We have now mentioned this point on page 5.

- The translocation times are quite long for such small molecules. Are the molecules expected to be in a stretched conformation, i.e. under tension, when inside the pore? As a rough reference, how do translocation times compare to the relaxation times?

Response 3.5: Yes, the translocation times are quite long in comparison with strong polyelectrolytes such as polystyrene sulfonate and DNA. We believe that this is due to (a) the net charge density being very small and (b) additional friction from significantly hydrated zwitterion moieties. In a separate project involving polyelectrolytes, we have attempted to determine the relaxation time using dynamic light scattering. As it turns out, it is not yet possible to make this determination primarily due to the emergence of a strong slow mode (corresponding to the spontaneous formation of dipole-driven aggregates). However, based on the Zimm dynamics model, the relaxation time is expected to be much shorter than the translocation time. Presently, due to the above-mentioned difficulties associated with the slow mode in dynamic

light scattering measurements, the relaxation times of these new class of polymers are yet to be experimentally determined.

- Given the above points and the low persistence length of the polymer, it is not clear that the polymer conformation is completely stretched out inside the pore. Even for the moderate “single-file” population, without being a blob the polymer conformation could easily be deflecting off the pore walls, i.e. not perfectly aligned with the pore axis. This locally angled conformation would result in deeper blockades, which might explain the 2.9 nm vs 2.2 nm difference. This would technically change the theoretical parameters of the models, but to first order wouldn't change conclusions.

Response 3.6: Please see the above Response 3.3 regarding the conformational fluctuations. Of course, the PSBMA is not fully stretched and there are conformational fluctuations (weak in the case of single-file moderate blockades and strong in the case of intra-pore-structured deep blockades). In terms of comparison between single-file experimental data and theory, the weak fluctuations are incorporated into the effective friction coefficient. While we can understand that the reviewer's comment arises primarily from their 2.9 nm vs. 2.2 nm difference, we don't think this difference occurs in realistic conditions due to hydration of zwitterionic moieties (please see Response 3.4). We agree with the reviewer that the conclusions of the paper will not change even if we were to possibly perform *ab initio* computations.

- The discussion regarding the folding interpretation is frankly a little vague and not visually supported by the data in the Figures. Where do the folded states occur and for how long? A complete understanding or quantitative description is not necessary, but a simple suggestion would be to add a figure of many (>20) current traces of translocation events to show the reader what events look like and show where and how long folded states are. These polymer translocations are not well known by the community, and a visual demonstration of signals would be very helpful in better understanding the data, and accepting the given interpretations.

Response 3.7: We thank the reviewer for this useful suggestion. We agree. We now have provided figures (Supplementary Figures 3 and 6) with many samples of ionic current traces for the intermediate and deep blockade events. We have also

enhanced the clarity of the discussion as mentioned in Responses 3.3 and 3.4.

- Unlike folded dsDNA blockades, blockage levels are far from being quantized ($\Delta I_1=0.6, \Delta I_2=0.8$). As discussed, this instead suggests local kinks in the polymer conformations, and not necessarily folds, which might not be too surprising given how flexible the polymer is with its 2 nm persistence length. Maybe the authors need to define folding (we assume it is a complete fold of the polymer chain on itself, thus producing quantized levels) and if they assume such kinks are folds (which we would instead define as in a slight blob state, but not fully stretched).

Response 3.8: We agree that the conformations have only local kinks and not such umbrella-handle-like folds observed in dsDNA blockades. Although this is what we meant by folded conformation in the revised version, we admit that this is potentially confusing given the large body of literature on folded dsDNA translocation. We have now replaced the word ‘folded’ by ‘intra-pore-structured’. Please see Response 3.3.

2. Lack of PMPC results.

- Using PMPC as a proof of the universality of CSB is great, however the analysis of PMPC results is too minimal and binary.

- As done with the PMBSA, a simple comment on the voltage-dependence of translocation time would demonstrate that polymers are actually translocating through the pore, as opposed to simply colliding with it. Such a comment was already made in a response to one of our comments but omitted from their revised manuscript.

Response 3.9: We agree. As we have already stated in the main text, such in-depth study of PMPC and other polyelectrolytes is of great interest to us and is relegated to future work. For the primary focus of CSB in the present manuscript, the observation of opposite polarity dependence of PMPC to that of PSBMA is sufficient to demonstrate CSB. However, to satisfy the reviewer’s curiosity, we would like to share our preliminary results given in the figure below showing that the frequency of translocation of PMPC increases with applied voltage. In addition, our data analysis shows that the average translocation time decreases from 0.55 ms at 100 mV to 0.41 ms at 200 mV. We are hesitant to present these preliminary results in the main text and we prefer to perform extensive and rigorous investigations on

PMPC and other polyzwitterions for future publications.

FIG. 2: Voltage dependence of capture rate of 100 nM PMPC in 3.6 M LiCl, 10 mM HEPES, pH 3.7. Pore diameter: 2.8 nm, sampling rate: 250 kHz, low-pass filter frequency: 10 kHz.

Minor points:

- 1. On page 5, the scale for r in [nm] in Figure 1g is confusing. Are the dielectric constant values reported for different distances? In that case it should be in [Angstroms]. h) is not introduced in the caption

Response 3.10: We thank the reviewer for this catch. Done.

- On page 7, PMPC is mentioned but has not been introduced or defined yet.

Response 3.11: Thanks for noticing this. Done.

- For overall readability: Subjectively, the introduction of the article “summarizes” the results of PMBSA in too much details, which makes for a repetitive read when going through the first pages of the results section (page 7).

Response 3.12. We agree to move the ‘summarizing paragraph’ in the Introduction to Results, as also suggested by the second reviewer.

- The population separation for moderate and deep populations is much better visualized with LiCl data (Figure 8 in SI) or 70 mV and 90 mV 1M KCl data sets (Figure 4 in SI). Why not use these in Figure 3c instead of the barely differentiated

populations of 150 mV? It would help the reader reach the same conclusions as the authors without making them dig and scroll through the SI.

Response 3.13: We have now presented data at 90 mV in Figure 3 as suggested by the reviewer.

Martin Charron & Vincent Tabard-Cossa

We sincerely thank the reviewer for their additional remarks to enhance the clarity of presentation and for their endorsement of the validity of CSB in the new materials class of neutral zwitterionic polymers.

Response to Reviewer #4

We thank the reviewer for devoting time and interest in reviewing this paper. As already mentioned in our earlier response, the review from the third reviewer (with whom the current reviewer has presumably collaborated) is of highest quality in terms of technical knowledge and scholarship. Thanks. The recent comment of the reviewer is in black and our response is in blue.

Response: As mentioned above, we thank you for your time and we appreciate the senior reviewer for providing this opportunity. The corresponding author of the present paper is sincerely committed to mentoring junior colleagues who are the future of the whole scientific endeavor.

Response to Reviewers #3 and #4

We thank the reviewers for the recommendation to publish the revised manuscript.

The comments of the reviewer are in black and our response is in blue.

Reviewer #3 (Remarks to the Author):

We thank the authors for addressing our latest sets of comments. We now support publication in Nature Communications. The added interpretation (text and schematics) of their polymer conformations will help the readers understand better the nature of the signals observed. We look forward to seeing more results on PMPC molecules and other polyzwitterion polymers in future studies.

Reviewer #4 (Remarks to the Author):

Response: We thank both reviewers for the recommendation and their contribution to enhance the clarity of the presentation of the key concept and data analysis behind CBS.

Response to Reviewer #2

The comments of the reviewer are in black and our brief response is in blue.

My comments after the second revision round remain largely unchanged because the authors have (once more) chosen to not perform the additional experiments I requested. Again, I do think this work has interesting aspects to it, but I also believe the results and conclusions are not solid and consistent enough to warrant publication in Nature Communications. I think we can respectfully agree to disagree that the mechanism is “clearly validated without any doubt”. I will give a few additional comments, but overall I feel like I’m repeating myself and I guess it will not change anything. I suppose the decision whether to publish the paper anyway is up to the editor.

In their previous reply, the authors argued that the CSB (and the electrophoretic mobility) is associated with the linear configuration of the chain enforced by the pore: “In bulk solutions, a dispersed flexible polymer such as PSBMA or PMPC is not a rigid sphere, a rigid cylinder, or a rigid ellipsoid . . . The net outcome is that these orientations and chain conformations are averaged out . . . Such a conformational status of the molecule prevents the distinction between the ionic groups on the repeat unit in terms of their normal distance from the chain backbone . . . Therefore, we designed our experiment by trying to transport the molecule through a nanopore as an unfolded single file.” After this I pointed out that the data suggests that the chains exhibit electrophoretic mobility also in the bulk solution due to the linear capture rate with voltage. Now it seems that the authors have changed their mind: “Of course, CSB is present even in bulk solution. Otherwise the neutral dipolar polyzwitterion would not move under an electric field to be captured by the nanopore . . .” It’s perfectly fine to change opinion, but (as I mentioned) this means that the whole interpretation of the results and the main point of the paper makes no sense. Based on the authors’ own description of the phenomenon, how can CSB occur for the randomly oriented and fluctuating chain? Why would it be captured by a field when in the coil state where all orientations are “averaged out”? The first step to be able to answer this critical question is to investigate by a simple independent method if there really is electrophoretic mobility in bulk or not.

Regarding my request to test zeta potentials, I agree that extraction of an accurate zeta potential value is not simple for a hydrated chain, but that is not the point. The measurements (PALS) can still be used to investigate field-induced mobility. The measurements are extremely simple to do by PALS and it should be known how the instruments calculates a zeta potential based on the measured mobility. I was also referring to streaming potential measurements on surface films. Again, these have revealed that organic materials believed to be neutral actually exhibit surface charges due to chemical interactions with ions. Another example: <https://doi.org/10.1021/jp004051u> There are also numerous examples of polyelectrolytes which exhibit unexpected charge behaviour depending on the electrolyte. (I'm not sure why the authors discuss issues with single molecule measurements here, there would be no need to perform them on the single molecule level.)

Finally, I point out once more that experiments on surface interactions between the polymers and silicon nitride would provide further important information. Again, these would be extremely simple to perform with established methods.

Response: (1) Averaging out conformations does not mean that the net charge must be zero, when there are local gradients in polarizability inside the macromolecule. (2) We have not changed any of our opinions in the course of the review process. Right in the first version of the manuscript, we discussed positive capture rate of PSBMA towards the positive electrode and zero capture rate towards the negative electrode. There can be only one opinion for this observation as already announced (title of the manuscript) in the original version. We simply had to repeat this concept in different words (and several times) with the hope of explaining the idea to this reviewer with apparently no success. (3) We hope that our paper will stimulate the reviewer and the general readership to conduct the reviewer's so-called "extremely simple" experiments on polyzwitterions.